# Dopamine enhances model-free credit assignment through boosting of retrospective model-based inference

**Lorenz Deserno**[1,2,3,4]*[†], **Rani Moran**[1,2]*[†], **Jochen Michely**[1,2,5], **Ying Lee**[1,2,4], **Peter Dayan**[1,6,7], **Raymond J Dolan**[1,2]

[1]Max Planck UCL Centre for Computational Psychiatry and Ageing Research, University College London, London, United Kingdom; [2]The Wellcome Trust Centre for Neuroimaging, Institute of Neurology, University College London, London, United Kingdom; [3]Department of Child and Adolescent Psychiatry, Psychotherapy and Psychosomatics, University of Würzburg, Würzburg, Germany; [4]Department of Psychiatry and Psychotherapy, Technische Universität Dresden, Dresden, Germany; [5]Department of Psychiatry and Psychotherapy, Charité Universitätsmedizin Berlin, Berlin, Germany; [6]Max Planck Institute for Biological Cybernetics, Tübingen, Germany; [7]University of Tübingen, Tübingen, Germany

**Abstract** Dopamine is implicated in representing model-free (MF) reward prediction errors a as well as influencing model-based (MB) credit assignment and choice. Putative cooperative interactions between MB and MF systems include a guidance of MF credit assignment by MB inference. Here, we used a double-blind, placebo-controlled, within-subjects design to test an hypothesis that enhancing dopamine levels boosts the guidance of MF credit assignment by MB inference. In line with this, we found that levodopa enhanced guidance of MF credit assignment by MB inference, without impacting MF and MB influences directly. This drug effect correlated negatively with a dopamine-dependent change in purely MB credit assignment, possibly reflecting a trade-off between these two MB components of behavioural control. Our findings of a dopamine boost in MB inference guidance of MF learning highlight a novel DA influence on MB-MF cooperative interactions.

**\*For correspondence:**
deserno_l@ukw.de (LD);
rani.moran@gmail.com (RM)

[†]These authors contributed equally to this work

**Competing interest:** The authors declare that no competing interests exist.

## Editor's evaluation

This behavioral pharmacology study provides convincing evidence that dopamine governs interactions between model-based and model-free learning systems. The issue addressed is timely and clinically relevant, and will be of interest to a broad audience interested in dopamine, learning, choice, and planning.

## Introduction

Dual system theories of reinforcement learning (RL) propose that behaviour is controlled by a prospective, model-based (MB), planning system and a retrospective, model-free (MF), value-caching system (*Daw and Dayan, 2014*; *Daw et al., 2005*; *Dolan and Dayan, 2013*). MF value-caching is driven by reward prediction errors (RPEs) that are signalled by phasic dopamine (*Montague et al., 1996*; *Schultz et al., 1997*; *Steinberg et al., 2013*), a finding from macaques and rodents, and mirrored in human neuroimaging studies (*D'Ardenne et al., 2008*; *O'Doherty et al., 2004*).

Early studies, motivated by classical psychological dual control distinctions regarding goal and habit (*Balleine and Dickinson, 1998*; *Dickinson, 1997*; *Dickinson, 1985 Dolan and Dayan, 2013*), focused on how these systems operated separately, with subsequent work indicating their influences being combined only close to the point of choice (*Daw et al., 2011*). However, even in the former and other experiments (involving the well-described two-step task), human functional neuroimaging showed that activity in the ventral striatum and the prefrontal cortex reflected MB values as well as MF RPEs (*Daw et al., 2011*; for replication, see *Deserno et al., 2015b*; *Deserno et al., 2015a*). More recent work has proposed a richer integration between MF and MB values (e.g., *Keramati et al., 2016*) as well as following an earlier influential suggestion as to how an MB system might train an MF system (*Mattar and Daw, 2018*; *Sutton, 1991*). More recent evidence has raised the possibility that this may occur during rest periods (*Antonov et al., 2021*; *Liu et al., 2021a*).

There has also been much interest in the influence of the neuromodulator DA in regulating these systems. The traditional assumption that DA RPEs are restricted to training MF values (referred to as MF credit assignment [MFCA]) has been challenged by a wealth of evidence that DA neuromodulation also impacts MB learning (MB credit assignment [MBCA]) and control (*Doll et al., 2012*; *Langdon et al., 2018*). Animal work suggests that DA RPEs reflect a hidden-state inference (*Starkweather et al., 2017*), while optogenetic activation and silencing of DA neurons impacts the efficacy of MB learning (*Sharpe et al., 2017*). Furthermore, the activity of DA neurons is reported to reflect the expression of MB values (*Sadacca et al., 2016*).

In humans, prior instructions about outcome probabilities (putatively reflecting an MB influence) exert control over MF learning and this effect correlates with genetic proxies of DA function (*Doll et al., 2009*; *Doll et al., 2012*). Boosting DA levels in the two-step task enhances MB influences over choice (*Sharp et al., 2016*; *Wunderlich et al., 2012*), while genetic or neurochemical measures of DA function reveal a positive association with MB influences (*Deserno et al., 2015a*; *Doll et al., 2016*). This positive relation between DA levels and MB influences has been replicated in a non-human animal study using an analogous paradigm (*Groman et al., 2019*). A multimodal imaging study in humans, using the two-step task, has also shown that inter-individual differences in ventral striatal DA differentially relates to the degree of the mutual representation of MF RPEs and MB values, in ventral striatum and prefrontal cortex, respectively (*Deserno et al., 2015a*).

However, the two-step task does not allow a ready dissociation of MB contributions to choice from the possibility that the MB system might train an MF system. The latter could, for instance, underlie the mutual representation of MF RPEs and MB values found in the ventral striatum (*Daw et al., 2011*; *Deserno et al., 2015b*; *Deserno et al., 2015a*) and, at least in part, account for DA's effect in boosting a net MB influence over choice. On this basis, *Moran et al., 2019*, designed a dual-outcome bandit task involving two different experimental manipulations so as to separate both MF and MB influences over choice, and a particular form of MB influence over MF learning. In order to examine the influence of boosting DA in our novel task, we conducted a double-blind, placebo-controlled, within-subjects pharmacological study, employing levodopa to boost overall DA levels whilst administering this novel dual-outcome bandit task. Based on previous work showing a positive influence of DA on MB control, we hypothesized that enhancing DA would strengthen an MB guidance of MFCA, as well as enhance MB influence over choice.

Foreshadowing our results, we found that boosting DA levels via levodopa enhanced MB guidance of MFCA. Unexpectedly, we did not find that boosting DA impacted MB influence over choice. Instead, we observed that the drug effects on MB guidance of MFCA and on MB influence over choice were negatively correlated.

## Results

### Study and task design

We conducted a placebo-controlled, double-blind, within-subjects pharmacological study using levodopa to enhance presynaptic DA levels, as in previous studies (*Chowdhury et al., 2013*; *Wunderlich et al., 2012*). Participants were tested twice, once under the influence of 150 mg levodopa, and once on placebo, where drug order was counterbalanced across individuals (n = 62, *Figure 1A*; cf. Materials and methods). On each lab visit, participants performed a task first introduced previously by *Moran et al., 2019*. The task was framed as a treasure hunt game called the 'Magic Castle'. Initially,

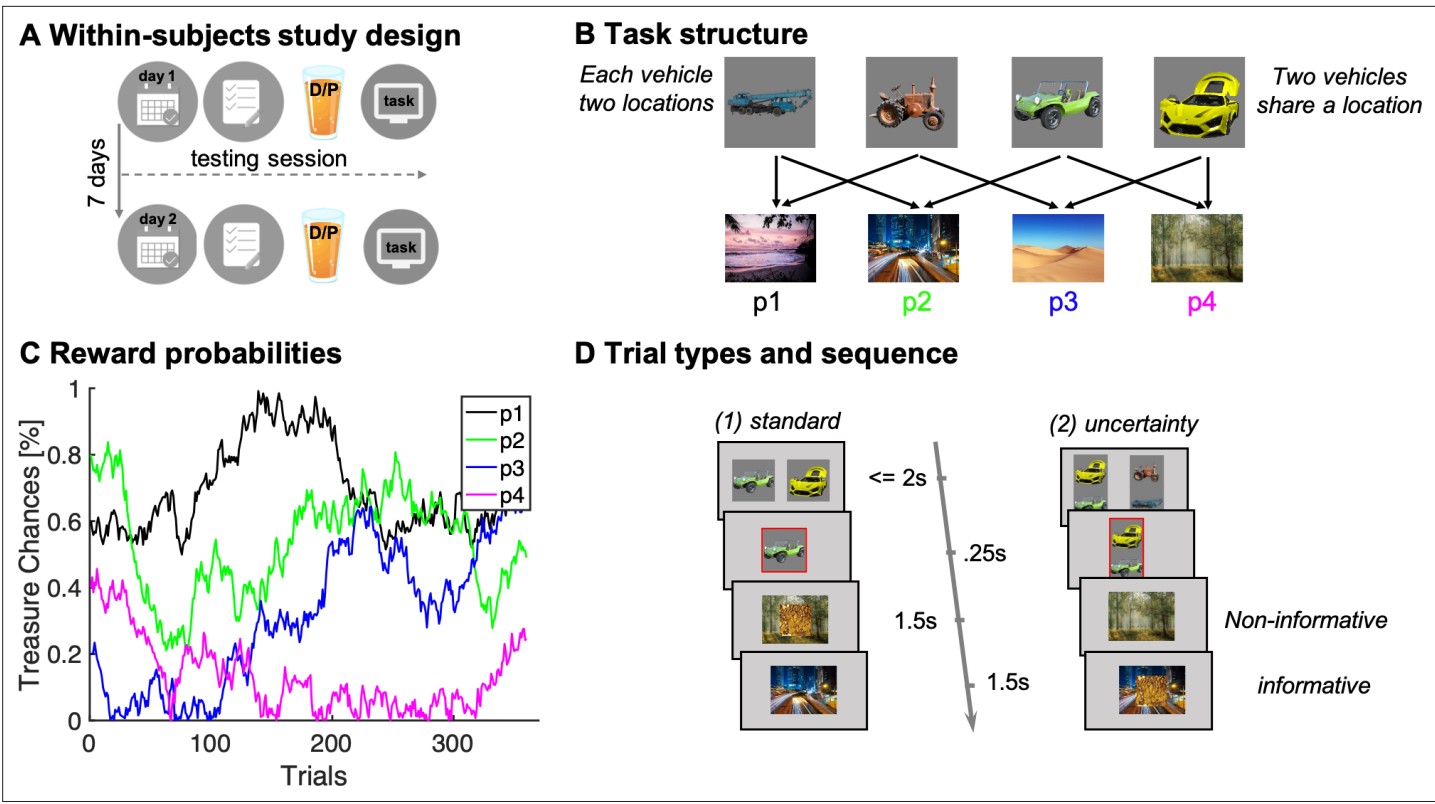

**Figure 1.** Study and task design. (**A**) Illustration of within-subjects design. On each of 2 testing days, approximately 7 days apart, participants started with either a medical screening and brief physical exam (day 1) or a working memory test (day 2). Subsequently they drank an orange squash containing either levodopa ("D") or placebo ("P"). (**B**) Task structure of the Magic Castle Game. Following a choice of vehicle, participants 'travelled' to two associated destinations. Each vehicle shared a destination with another vehicle. At each destination, participants could win a reward (10 pence) with a probability that drifted slowly as Gaussian random walks, illustrated in (**C**). (**D**) Depiction of trial types and sequences. (1) On *standard* trials (2/3 of the trials), participants made a choice out of two options in trial n (max. choice 2 s). The choice was then highlighted (0.25 s) and participants subsequently visited each destination (0.5 s displayed alone). Reward, if obtained, was overlaid to each of the destinations for 1 s. (2) On *uncertainty trials*, participants made a choice between two pairs of vehicles. Subsequently, the ghost nominates, unbeknown to the participant, one vehicle out of the chosen pair. Firstly, the participant is presented the destination shared by the chosen pair of vehicles (here the forest) and this destination is therefore *non-informative* about the ghost's nominee. Secondly, the destination unique to the ghost-nominated vehicle is then shown (the highway). This second destination is *informative* because it enables inference of the ghost's nominee with perfect certainty based on a model-based (MB) inference that relies on task transition structure. Trial timing was identical for standard and uncertainty trials.

participants were trained extensively on a transition structure between states, under a cover narrative of four vehicles and four destinations. Subjects learned that each vehicle (state) travelled to two different sequential destinations in a random order (*Figure 1B*). The mapping of vehicles and destinations remained stationary throughout a session, but the two test sessions (drug/placebo) featured different vehicles and destinations. At each destination, participants could potentially earn a reward with a probability that drifted across trials according to four independent random walks (*Figure 1C*).

The task included two trial types (*Figure 1D*): (1) standard trials (2/3 of the trials) and (2) uncertainty trials (1/3 of the trials). On standard trials, participants were offered two vehicles and upon choosing one, they visited both its associated destinations where they could earn rewards. On uncertainty trials, participants likewise chose a pair of vehicles (from two offered vehicle pairs). Next, an unseen ghost randomly nominated a choice of one of the vehicles in the chosen pair, and a visit to its two destinations followed. Critically, participants were not privy to which vehicle was nominated by the ghost. However, they could resolve this uncertainty after seeing both visited destinations based on their knowledge of the transition structure. We refer to this as retrospective MB inference. Such inference can only occur after exposure to the second destination, as only then can subjects know which of the two vehicles the ghost had originally selected.

## Logic of the dissociation between MFCA and MBCA

Following the approach adopted by *Moran et al., 2019*, an MF system relies solely on observations of earned rewards associated with a chosen vehicle in order to update its current value. By contrast, the MB system does not maintain and update values for the vehicles directly. Instead, the MB system updates values of destinations by calculating on-demand values prospectively for each vehicle offered for choice (see computational modelling). In effect, an MF system updates values only for the chosen vehicle (e.g., for the green antique car based on the sum of rewards at the forest and highway as illustrated in *Figure 2A and B1*). An MB system can generalize across vehicles, which share a common destination (e.g., after choice of the green antique car, values for the forest and highway are updated; this enables choice generalization to the yellow racing car after a reward at the forest as illustrated in *Figure 2A & C1*).

In uncertainty trials, without guidance from an MB system, a pure MF system cannot assign credit preferentially to the ghost-nominated vs. the ghost-rejected vehicle value. This is because the ghost's nomination is covert (e.g., MF cannot infer the ghost's choice of the green antique car, *Figure 3*). However, an MB system can infer which vehicle the ghost nominated at the second, informative destination (the highway in *Figure 3*), and then, at least in principle, guide the MF system to assign credit purely to that vehicle. We use characteristics of choice on subsequent standard trials n + 1 (illustrated in *Figures 3 and 4*) to infer whether there has indeed been MFCA to that vehicle. By contrast, based on purely MB evaluations, an MB system can update the value of each destination as on sequences of standard trials. Note that guidance of MFCA and pure MBCA should therefore be considered as two different MB processes. Previous studies on the contribution of DA to the operation of MF and MB systems (*Deserno et al., 2015b*; *Doll et al., 2016*; *Kroemer et al., 2019*; *Sharp et al., 2016*; *Wunderlich et al., 2012*) could not dissociate between such distinct MB processes.

We first present 'model-agnostic' analyses focusing on how events on trial n affect choices on trial n + 1. This allows identification of MF and MB choice signatures (standard trials n followed by standard trials n + 1), the guidance of MFCA by retrospective MB inference (uncertainty trials n followed by standard trials n + 1), and, crucially, whether these signatures varied as a function of drug treatment (levodopa vs. placebo). The model-agnostic analyses are also supplemented by simulations and data fitting based on computational models.

## No evidence of dopaminergic modulation for MF choice

The MF system updates values of chosen vehicles alone based on the sum of earned rewards at each associated destination. This is revealed as a tendency to repeat choices. As illustration, consider a sequence of standard trials n and n + 1 for which the vehicle chosen on the former (the green antique car) is also offered on the latter, against another vehicle (the yellow racing car *Figure 2 B1*). The two vehicles offered on trial n + 1 reach a common destination (the forest), but the vehicle previously chosen on trial n (the green antique car) also visits a unique destination. Because the MB systems know the common destination (the forest) is equally reachable, no matter what the choice, its value cancels out from MB calculations during the trial n + 1 choice. By contrast, the MF tendency to repeat the chosen vehicle, the green antique car, at trial n + 1 will increase or decrease respectively, when the common destination is or is not rewarded on trial n.

In a binomial mixed effects model (see Materials and methods and *Appendix 1—table 1*), we regressed a choice repetition of this vehicle on whether the common and/or unique destinations were rewarded (reward/non-reward) on trial n and on drug status (levodopa/placebo). Replicating a previous finding (*Moran et al., 2019*), we found a main effect for common reward (b = 0.67, t(480) = 9.14, p < 0.001). This effect constitutes MF choice repetition, as the MB system appraises that the common destination favours both trial n + 1 vehicles (see *Appendix 1—figure 1* for validating simulations). As expected on both MB and MF grounds, there was a main effect for unique reward (b = 1.54, t(480) = 17.40, p < 0.001). There was no drug × common reward interaction (b = 0.07, t(480) = 0.68, p = 0.500), providing no evidence for a drug-induced change in MF choice repetition on standard trials (*Figure 2 B2*). None of the remaining (main or interaction) effects were significant (*Appendix 1—table 1*).

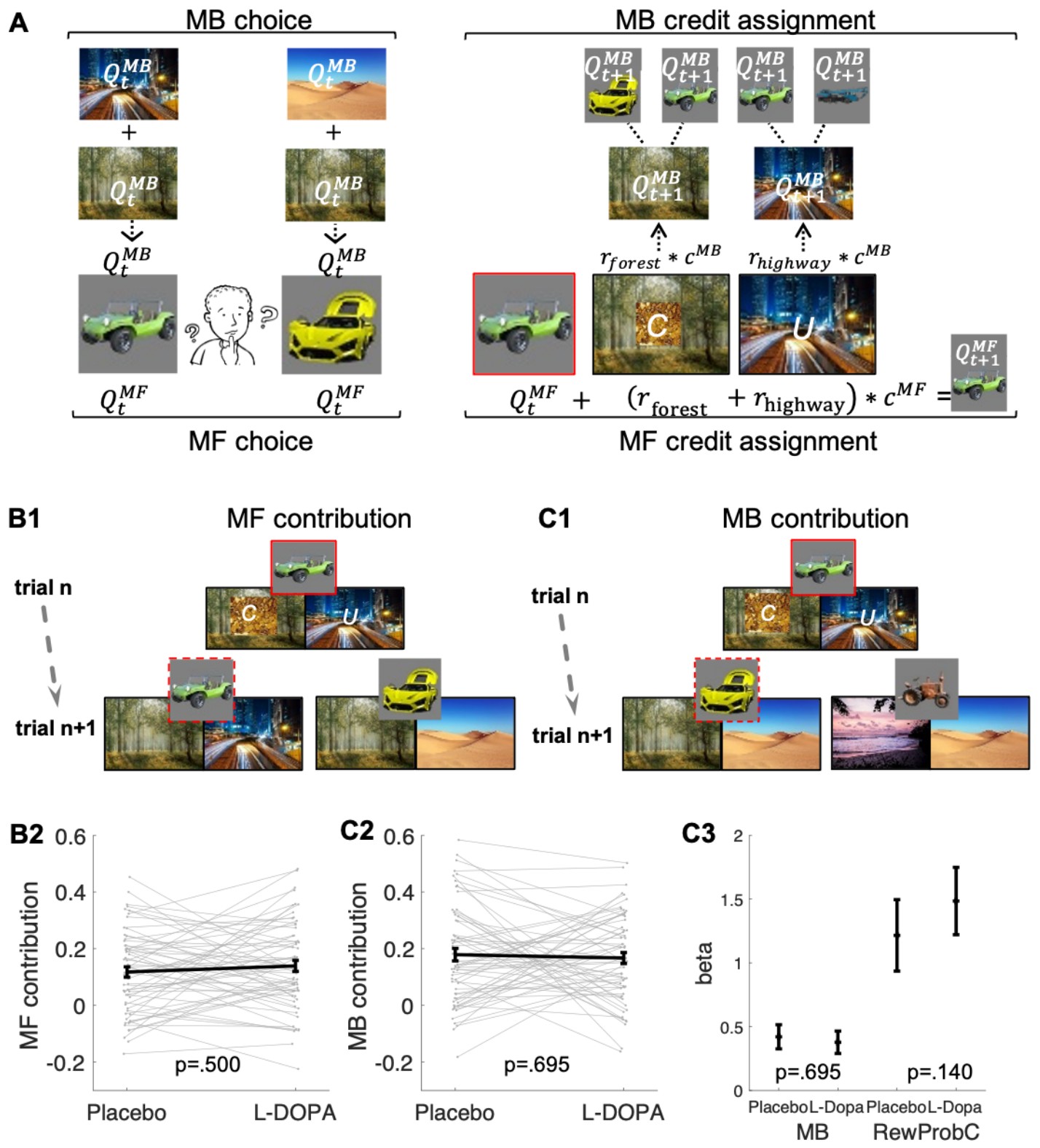

**Figure 2.** Model-free (MF) and model-based (MB) contributions. (**A**) Left panel: Illustration of MF and MB values at choice in standard trials. MB values are computed prospectively based on the sum of values of the two destinations associated with each of the two vehicles offered for choice (here highway and forest for green antique car; desert and forest for yellow racing car). Right panel: MF vs. MB credit assignment (MFCA vs. MBCA) in standard trials. MFCA only updates the chosen vehicle (here green antique car) based on the sum of rewards at each destination (here forest and highway). MBCA updates separately the values for each of the two destinations (forest and highway). Each of these updates will prospectively affect

*Figure 2 continued on next page*

*Figure 2 continued*

equally the values of the vehicle pair associated with that destination (updated MB value for forest influences MB value of the yellow racing car and the green antique car while the updated MB value for highway influences the MB value of the green antique car and the blue crane). Forgetting of Q values was left out for simplicity (see Materials and methods and see *Appendix 1—figure 1* for validating simulations). (**B1**) Illustration of MF choice repetition. We consider only standard trials n + 1 that offer for choice the standard trial n chosen vehicle (e.g., green antique car) alongside another vehicle (e.g., yellow racing car), sharing a common destination (forest). Following choice of a vehicle in trial n (framed in red, here the green antique car), participants visited two destinations of which one can be labelled on trial n + 1 as common to both offered vehicles (C, e.g., forest, which was also rewarded in the example) and the other labelled as unique (U, e.g., city highway, unrewarded in this example) to the vehicle chosen on trial n (the green antique car). The trial n common destination reward effect on the probability to repeat the previously chosen vehicle (dashed frame in red, e.g., the green antique car) constitutes an MF choice repetition. (**B2**) The empirical reward effect for the common destination (i.e., the difference between rewarded and unrewarded on trial n, see *Appendix 1—figure 2* for a more detailed plot of this effect) on repetition probability in trial n + 1 is plotted for placebo and levodopa (L-Dopa) conditions. There was a positive common reward main effect and this reward effect did not differ significantly between placebo and levodopa conditions. (**C1**) Illustration of the MB contribution. We considered only standard trials n + 1 that excluded from the choice set the standard trial n chosen vehicle (e.g., green antique car, framed in red). One of the vehicles offered on trial n + 1 shared one destination in common with the trial n chosen vehicle (e.g., yellow racing car, sharing the forest, and we term its choice a generalization). A reward (on trial n) effect for the common destination on the probability to generalize on trial n + 1 (e.g., by choice of the yellow racing car, dashed frame in red) constitutes a signature of MB choice generalization. (**C2**) The empirical reward effect at the common destination (i.e., the difference between rewarded and unrewarded, see *Appendix 1—figure 2* for a more detailed plot of this effect) on generalization probability is plotted for placebo and levodopa conditions. (**C3**) In the regression analysis described in the text, we also include the current (subject- and trial-specific) state of the drifting reward probabilities (at the common destination) because we previously found this was necessary to control for temporal auto correlations in rewards (*Moran et al., 2019*). For completeness, we plot beta regression weights of reward vs. no reward at the common destination (indicated as MB) and for the common reward probability (RewProbC) each for placebo and levodopa conditions. No significant interaction with drug session was observed. Error bars correspond to SEM reflecting variability between participants.

## No evidence of dopaminergic modulation of MB choice

The MB system updates destination values based on rewards and computes on-demand values for the offered vehicles prospectively. Thus, a reward for a destination will equally affect an MB tendency to choose any vehicle leading to that destination (whether the reward followed a choice of that vehicle or a different one). For illustration, consider a sequence of standard trials n and n + 1, which excludes the vehicle chosen on trial n (green antique car) from the choice set. The trial n chosen vehicle has a destination in common (the forest, annotated as C in *Figure 2C1*) with one of the trial n+1 offered vehicles (the yellow racing car, *Figure 2 C1*). Because the MB system knows the vehicle-destination mappings, a reward associated with the forest will tend to make it favour the yellow racing car on trial n+1, even though this car was not present on trial n. In contrast, the MF system assigns credit from the common destination trial n reward (or non-reward) only to the vehicle (the green antique car) that was chosen on that trial (and which is absent on trial n + 1). Hence, a reward at the common destination at trial n should not affect MF choice on trial n.

Using a logistic mixed effects model (see Materials and methods and *Appendix 1—table 1*), we regressed choice generalization on trial n rewards at the common destination, on the current reward probability of the common destination and on drug session, which replicated our previous finding (*Moran et al., 2019*) of a positive main effect for the common reward (b = 0.40, t(7177) = 6.22, p < 0.001). This positive common trial n reward effect on choice constitutes an MB choice generalization (even after controlling for the drifting reward probability at the common destination, see *Appendix 1—figure 1* for validating simulations). The common reward × drug interaction was not significant (b = 0.05, t(7177) = 0.39, p = 0.695), providing no evidence for a drug-induced change in the influence of MB choice (*Figure 2C2 & C3*). Except for the main effect of the drifting reward probability at the common destination, no other effects were significant (*Appendix 1—table 1*).

In summary, we replicate previous findings (*Moran et al., 2019*) of mutual MF and MB contributions to choices. There was no evidence, however, that these contributions were modulated by levodopa.

## Retrospective MB inference guides MFCA

We next addressed our main question: Does levodopa administration boost an MB guidance of MFCA through a retrospective MB inference? In an uncertainty trial, participants choose one out of the two pairs of vehicles (*Figure 1D*). Next, a ghost randomly nominates a vehicle from the chosen pair. Participants then observe both destinations (related to the nominated vehicle) accompanied by their rewards (see *Figure 3* for an illustration of MFCA and MBCA in uncertainty trials). Thus, left to its own

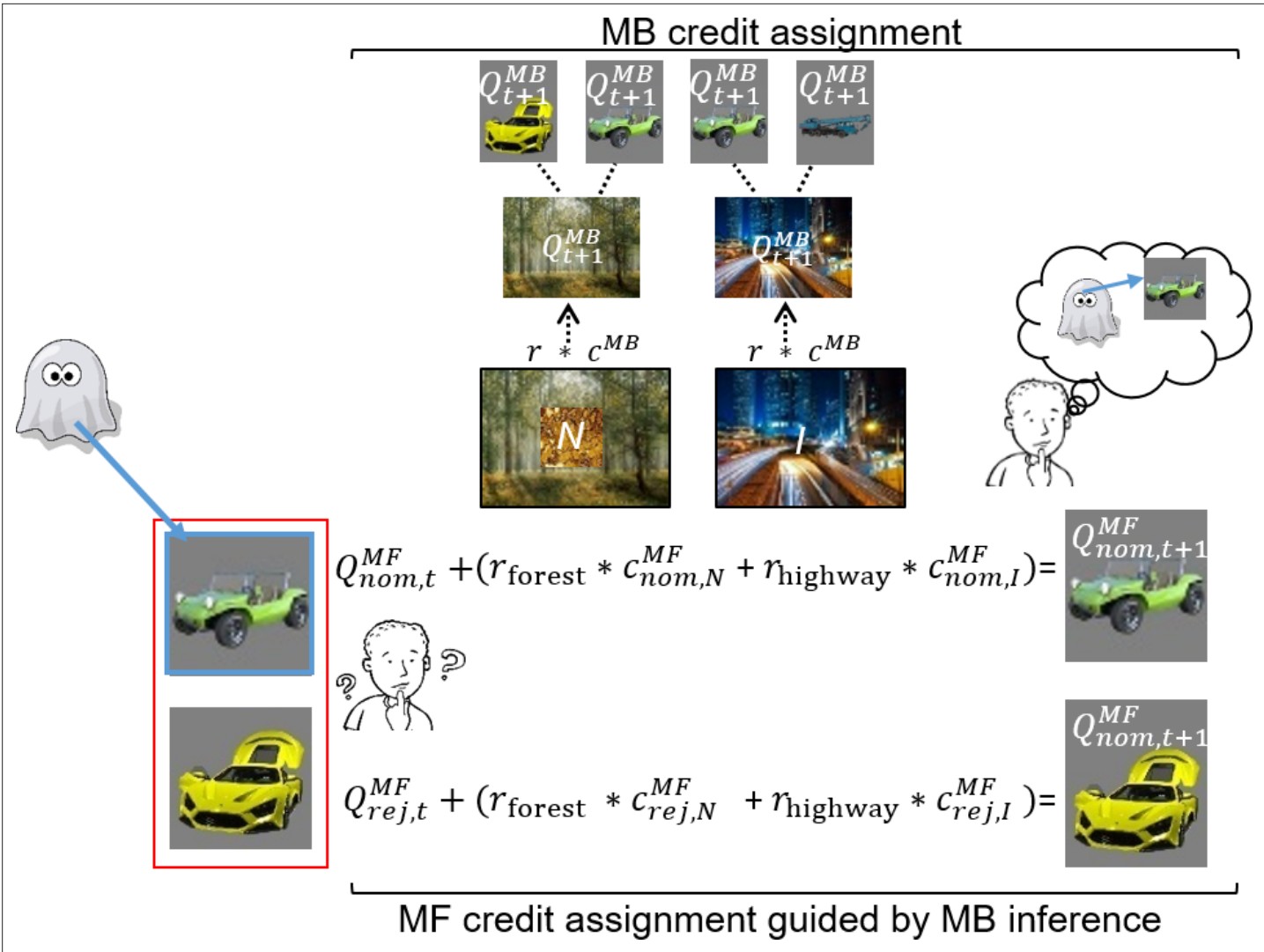

**Figure 3.** Illustration of model-free (MF) credit assignment (MFCA) guided by model-based (MB) inference and MB credit assignment (MBCA) in uncertainty trials. The ghost, unbeknown to the participants, nominates a vehicle (e.g., green antique car). The ghost's nomination does not matter for MBCA because it updates values for each of the destinations (here forest and highway) separately, which will prospectively effect on all associated vehicle values (here green antique car, yellow racing car, and blue crane). With respect to MFCA guided by MB inference, participants are in state uncertainty and have a chance belief about the ghost-nominated vehicle. The firstly presented destination (the forest) holds no information about the ghost-nominated vehicle (the green antique car), the non-informative ('N') destination. Thus, participants remain in state uncertainty. The destination presented second (here the highway) enables retrospective MB inference about the ghost's nomination (the green antique car) and is therefore informative ('I'). This retrospective MB inference enables preferential MFCA for the ghost-nominated vehicle (here green antique car) based on the sum of rewards at each destination (without such inference MFCA can only occur equally for the ghost-nominated and -rejected vehicles). Forgetting of Q values was left out for simplicity (see Materials and methods and see **Appendix 1—figure 3** for validating simulations).

devices, a pure MF system could update both vehicles equally; but because it does not know which of the two vehicles had been nominated by the ghost, it cannot update that one more. Equally, the MB system only updates the values of destinations, and so is not affected by the nomination process. Crucially, though, since the MB system works by knowing which destinations are associated with which vehicle, it could infer the ghost's choice, and then inform the MF system.

For illustration, when visiting destinations and their momentary rewards, participants first see a destination common to both of the vehicles of the chosen pair, followed by a destination unique to the ghost-nominated vehicle. As participants are uninformed about the ghost's nomination, they have a 50–50% belief initially and observing the first destination is *non-informative* with respect to the ghost's nominee (the forest in *Figure 3* as it is common to the vehicles, the green antique car and

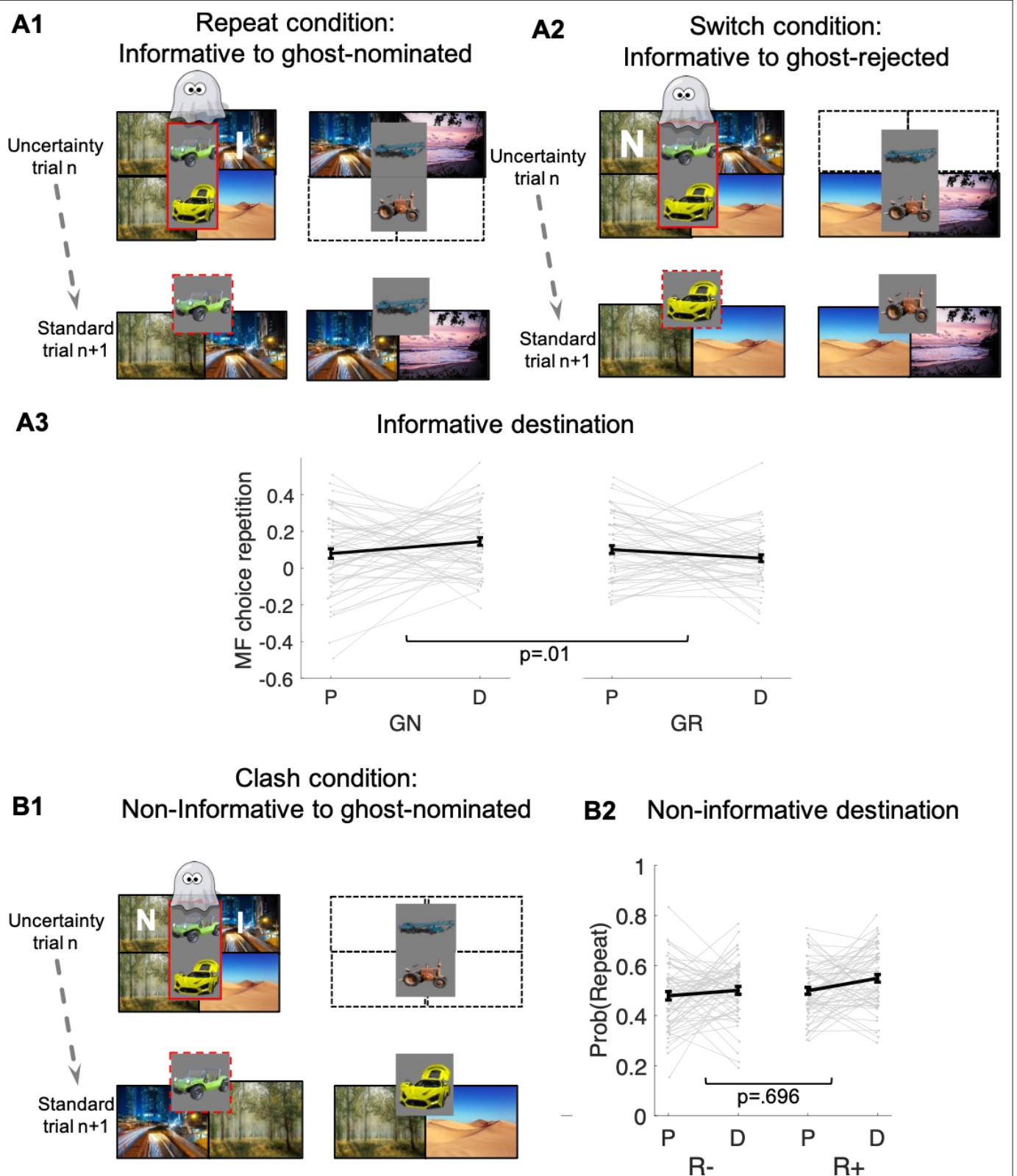

**Figure 4.** Guidance of model-free (MF) credit-assignment (CA) by retrospective model-based (MB) inference. (**A1**) Illustration of the repeat condition. The ghost-nominated vehicle (e.g., green antique car) is offered for choice in standard trial n + 1 alongside a vehicle from the non-chosen pair (e.g., blue building crane). A higher probability to repeat the ghost-nominated vehicle in standard trial n + 1 after a reward as compared to no reward at the informative destination, the highway, constitutes model-free credit assignment (MFCA) for the ghost's nomination (GN, the green antique car).

*Figure 4 continued on next page*

*Figure 4 continued*

(**A2**) Illustration of the switch condition. The ghost-rejected vehicle (e.g., the yellow racing car) is offered for choice in standard trial n + 1 alongside a vehicle from the non-chosen pair (e.g., brown farming tractor). A higher probability to choose the ghost-rejected vehicle in standard trial n + 1 after a reward as compared to no reward at the informative destination constitutes MFCA for the ghost's rejection (GR). Both ghost-based assignments depend on retrospective model-based (MB) inference. (**A3**) Preferential effect of retrospective MB inference on MFCA (effects of GN > GR) based on the informative destination is enhanced under levodopa (L-Dopa; "D") as compared to placebo ("P"). This is indicated by a significant trial type (GN/GR) × drug (placebo/ levodopa) interaction (also see *Appendix 1—figure 4* and *Appendix 1—figure 5* for more detailed plots). (**B1**) Illustration of the clash condition. The previously chosen pair (green antique and yellow racing car) is offered for choice in standard trial n + 1. A higher probability to repeat the ghost-nominated vehicle (the green antique car) in standard trial n + 1 following reward (relative to non-reward) at the non-informative destination (the forest) constitutes a signature of preferential MFCA for GN (the green antique car) over GR (the yellow racing car). (**B2**) Choice repetition in clash trial is plotted as a function of L-Dopa ("D") vs. placebo ("P") and reward (R+: reward; R−: no-reward, see *Appendix 1—figure 6* for a more detailed plot). While there was a main effect for drug, there was no interaction of non-informative reward × drug, providing no evidence that drug modulated MFCA based on the non-informative outcome. Error bars correspond to SEM reflecting variability between participants.

the yellow racing car). Critically, following observation of the second destination (e.g., the highway), an MB system can infer the ghost-nominated vehicle (the green antique car) with absolute certainty based upon knowledge of the task transition structure. Thus, the second destination, the highway, is retrospectively *informative* with respect to MB inference of the ghost's nominee. Subsequently, the inferred vehicle information can be shared with an MF system to direct MFCA towards the ghost-nominated vehicle (the green antique car). We assume guidance of MFCA occurs for both vehicles in the chosen pair, but to a different extent. Specifically, guidance of MFCA for the ghost-nominated (green antique car), as compared to the ghost-rejected (yellow racing car), vehicle would support an hypothesis that retrospective MB inference preferentially guides MFCA (*Moran et al., 2019*). See *Appendix 1—figure 3* for validating model simulations. Our novel hypothesis here is that this effect will be strengthened under levodopa as compared to placebo, which we examine, firstly via the informative destination, and, secondly via the non-informative destination.

## DA enhances preferential guidance of MFCA at the informative destination

MFCA at the informative destination to the ghost-nominated vehicle is tested by comparing 'repeat' and 'switch' standard trials n + 1 that follow an uncertainty trial n, as depicted in *Figure 4A1-A2*. In a 'repeat' trial (*Figure 4 A1*), the vehicle chosen by the ghost (the green antique car) is pitted against another vehicle from the non-chosen uncertainty trial n pair (the blue crane), which shares the informative destination (the highway). In a 'switch' trial, the vehicle the ghost rejected (the yellow racing car) is presented against a vehicle from the non-chosen pair (the brown tractor) that, likewise, does not share the informative destination. Consider what happens on trial n + 1 if this informative destination is rewarded on trial n. On a 'repeat' trial, the MB system knows that both vehicles on trial n + 1 lead to this destination, and so the reward on trial n is irrelevant to its choice. On a 'switch' trial (*Figure 4 A2*), the MB system knows that neither vehicle on trial n + 1 leads to this destination, and so the reward on trial n is again irrelevant to its choice. The MF system would normally assign credit equally to both vehicles on trial n, because it does not know the ghost's nomination, and so the green antique car on the 'repeat' trial and the yellow racing car on the 'switch' trial should benefit equally from a reward on the highway. However, if the MB system can exert an influence on MFCA then reward on the highway at trial n will favour choosing the green antique car (that the ghost actually nominated) on a 'repeat' trial, more so than it will favour choosing the yellow racing car (that the ghost did not nominate) on a 'switch' trial. Note that this guidance of preferential MFCA (PMFCA) can only take place once state uncertainty is resolved and therefore by definition can only be retrospective.

The key metric of interest from 'repeat' vs. 'switch' standard trials n + 1 for our drug analysis is the contrast between MFCA to ghost-nominated vs. ghost-rejected vehicles based on the reward effects at the informative destination. For repeat trials, we defined reward effects as the contrast between the proportion of choices (trial n + 1) of the vehicle the ghost nominated (on the uncertainty trial n) when the informative destination was previously rewarded vs. unrewarded. For switch trials, reward effects were defined similarly but now with respect to choices of the vehicle the ghost rejected (on the uncertainty trial n). We calculated reward effects separately (*Figure 4 A3*) for each type of standard trial n + 1 (repeat/switch; labelled as nomination) and for drug condition (levodopa/placebo). In a linear mixed

effects model (see Materials and methods and *Appendix 1—table 2*) where we regressed these reward effects, we found no main effect either of nomination (b = 0.03 t(239) = 1.57, p = 0.117) or of drug (b = 0.01, t(239) = 0.40, p = 0.687). Crucially, we found a significant nomination × drug interaction (b = 0.11, t(239) = 2.73, p = 0.007). A simple effects analysis revealed a PMFCA of the ghost-nominated over the ghost-rejected vehicle was significant under levodopa (b = 0.09, F(239,1) = 9.07, p = 0.003) but not under placebo (b = −0.02, F(239,1) = 0.52, p = 0.474). This supports our hypothesis that levodopa preferentially enhanced MFCA for the ghost-nominated, compared to ghost-rejected, vehicle under the guidance of retrospective MB inference. The nomination × drug interaction was not affected by session order (see *Appendix 1—table 2*).

## Dopaminergic modulation of PMFCA at the non-informative destination

A second means to examine MB influences over MFCA is to consider the non-informative destination. We refer to this as the clash condition (*Figure 4 B1*). Here, we only analyze standard trials n + 1 that offer for choice the ghost-nominated (the green antique car) against the ghost-rejected vehicle (the yellow racing car). Importantly, all such trials are included in the analysis irrespective of whether rewards were administered at the informative (the highway) and/or non-informative destination (the forest) or not. Either type of reward has no impact on a pure MF (i.e., unguided by MB) system's choice on trial n + 1, since it assigns these rewards equally to both ghost-nominated and ghost-rejected vehicles on trial n. Thus, we need only consider MB choice and MB guidance of MFCA and MF choice on trial n + 1. First, consider the effect of reward at the highway. This favours the green antique car either directly by MB choice on trial n + 1 or by MB guidance of MFCA and MF choice on trial n + 1. Thus, any effect of reward at the informative destination is confounded. By contrast, consider a reward to the non-informative destination (the forest). Because this destination is shared between the green antique car and the yellow racing car, an MB choice on trial n + 1 does not favour either vehicle. However, if an MB system has guided an MFCA on trial n to the green antique car, and the MF system controlled choice on trial n + 1, then a choice of the green antique car should be favoured. Note again that this guidance can only be retrospective, since it is only when the highway is presented on trial n that the MB system can infer that the ghost had in fact nominated the green antique car.

We previously showed that a positive effect of reward at the non-informative destination on choice repetition (i.e., a choice of the previously ghost-nominated vehicle), implicating a preferential guidance of MFCA towards the ghost-nominated vehicle guided by retrospective MB inference (*Moran et al., 2019*). Here, in a binomial mixed effects model (see Materials and methods and *Appendix 1—table 2*), we regressed choice repetition on trial n rewards at informative and non-informative destinations as well as on drug session. A marginally significant main effect for a reward at the non-informative destination provides trend level support for PMFCA to the ghost-nominated vehicle (b = 0.13, t(479) = 1.96, p = 0.051). Additionally, we found a main effect for reward at the informative destination (b = 1.01, t(479) = 9.95, p < 0.001), as predicted by both the enhanced MFCA for the ghost-nominated vehicle and by an MB contribution. The interaction effect between drug and non-informative reward, however, was not significant (b = 0.05, t(479) = 0.39, p = 0.696, *Figure 4B2*), nor were any other interactions in the model (*Appendix 1—table 2*). Thus, this analysis yielded no evidence that levodopa enhanced preferential guidance of MFCA to the ghost-nominated vehicle based on reward at a non-informative destination. Unexpectedly, we found a positive main effect of drug (b = 0.15, t(479) = 2.31, p = 0.021, *Figure 4B2*), indicating that participants' tendency to repeat choices of the ghost-nominated vehicle was enhanced under levodopa. This finding was only seen in this specific subset of trials and did not emerge in the computational modelling we now report below.

## Computational modelling

One limitation of the analyses reported above is that they isolate the effects of the immediately preceding trial on a current choice. However, values and actions of RL agents are influenced by an entire task history and, to take account of such extended effects, we formulated a computational model that specified the likelihood of choices (*Moran et al., 2019*, also see *Moran et al., 2021a*; *Moran et al., 2021b*; *Moran et al., 2021c*). We address differences in the modelling approaches in detail in SI. In brief, at choice, MF values ($Q^{MF}$) of the two presented vehicles feed into a decision module. During learning, the MF system updates $Q^{MF}$ of the *chosen* vehicle (the green antique car in *Figure 2A*) based on the sum of earned rewards from each destination (forest and highway in

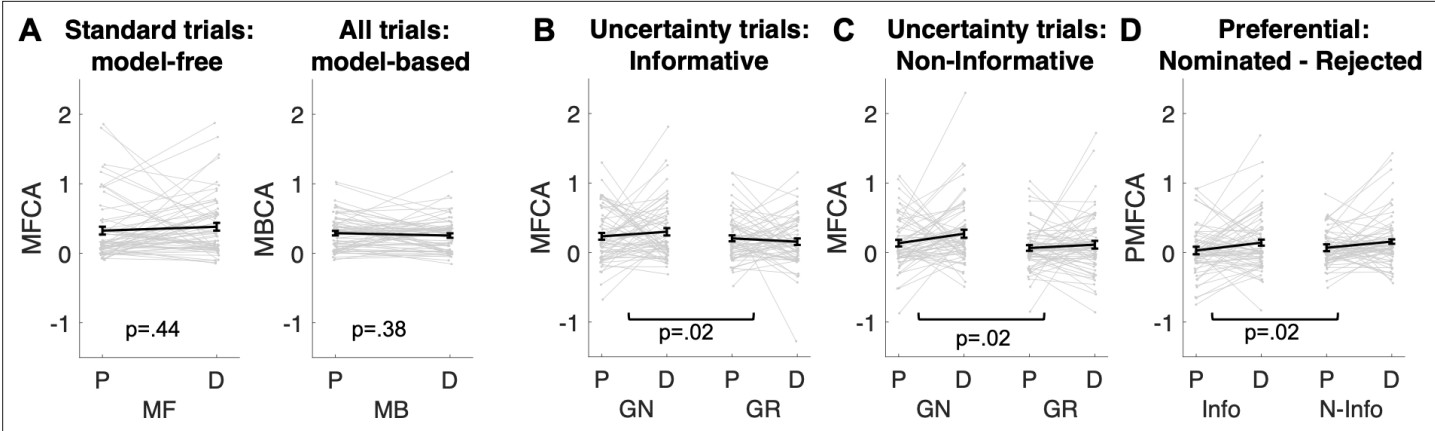

**Figure 5.** Analyses based on estimated credit assignment (CA) parameters from computational modelling (for model comparisons, based on the current and the *Moran et al., 2019*) data, see *Appendix 1—figure 7* and *Appendix 1—figure 8*; for parameter recoverability, see *Appendix 1—figure 9*. (**A**) Model-free and model-based credit assignment parameters (MFCA; MBCA) did not differ significantly for placebo (**P**) and levodopa (**D**) conditions. (**B**) MFCA parameters based on the informative destination for the ghost-nominated (GN) and the ghost-rejected (GR) destinations as a function of drug condition. (**C**) Same as B but for the non-informative destination. (**D**) The extent to which MFCA prefers the ghost-nominated over the ghost-rejected vehicle for each destination and drug condition. We name this preferential MFCA (PMFCA). Error bars correspond to SEM reflecting variability between participants.

*Figure 2A*). By contrast, the MB system prospectively calculates on-demand $Q^{MB}$ values for each offered vehicle (green antique and yellow racing car in *Figure 2A*) based on an arithmetic sum of the values of its two destinations (forest and highway for green antique car and forest and desert for yellow racing car):

$$Q^{MB}\left(\text{vehicle}\right) = Q^{MB}\left(\text{correspondingdestination1}\right) + Q^{MB}\left(\text{correspondingdestination2}\right) \quad (1)$$

During learning, the MB system updates the values of the two visited destinations (for forest and highway after choice of the green antique car, *Figure 2A*). We refer to these updates as MBCA. Unlike MFCA, which does not generalize credit from one vehicle to another, MBCA generalizes across the two vehicles which share a common destination. Thus, when a reward is collected in the forest destination, $Q^{MB}$ (forest) increases. As the forest is a shared destination, both vehicles that lead to this destination benefit during ensuing calculations of the on-demand $Q^{MB}$ values. Critically, our model included five free 'MFCA parameters' of focal interest, quantifying the extent of MFCA on standard trials (one parameter), on uncertainty trials (four parameters) for each of the objects in the chosen pair (nominated/rejected), and for each destination (informative/non-informative). We verified that the inclusion of these parameters was warranted using systematic model comparisons. A description of the sub-models and the model selection procedure is reported in the Materials and methods section and in *Appendix 1—figure 7*. Notably, the validity of our current novel model is also supported in a reanalysis of data from *Moran et al., 2019* (see *Appendix 1—figure 8*). We fitted our full model to each participant's data in drug and placebo sessions based on maximum likelihood estimation (see Materials and methods). Parameter recoverability is reported in Appendix 1 (*Appendix 1—figure 9*).

## Absence of dopaminergic modulation for MBCA and MFCA

In line with our model-agnostic analyses of standard trials, we found positive contributions of MFCA (parameter $c_{standard}^{MF}$; *Figure 5A*) for both levodopa (M = 0.381, t(61) = 6.84, p < 0.001) and placebo (M = 0.326, t(61) = 5.76 p < 0.001), with no difference between drug conditions (t(61) = –0.78, p = 0.442). Likewise, MBCA (parameter $c^{MB}$; *Figure 5A*) contributed positively for both levodopa (M = 0.255, t(61) = 7.88, p < 0.001) and placebo (M = 0.29, t(61) = 8.88, p < 0.001), with no significant difference between drugs (t(61) = 0.88, p = 0.3838). Thus, while both MBCA and MFCA contribute to choice, there was no evidence for a drug-related modulation. Forgetting and perseveration parameters of the model did not differ as a function of drug (see Appendix 1).

## Levodopa enhances guidance of PMFCA by retrospective MB inference on uncertainty trials

To test our key hypothesis, that guidance of PMFCA by retrospective MB inference on uncertainty trials is enhanced by levodopa, we focused on the four computational parameters that pertaining to MFCA on uncertainty trials ($c^{MF}_{nom,info}, c^{MF}_{rej,info}, c^{MF}_{nom,noninfo}, c^{MF}_{rej,noninfo}$, *Figure 5B and C*). In a mixed effects model (see Materials and methods and *Appendix 1—table 3*), we regressed these MFCA parameters on their underlying features: nomination (nominated/rejected), informativeness (informative/non-informative), and drug session (levodopa/placebo). Crucially, we found a positive nomination × drug interaction (b = 0.10, t(480) = 2.43, p = 0.015). A simple effects analysis revealed PMFCA (the effect of nomination) to be significant under levodopa (b = 0.13, F(480,1) = 9.01, p = 0.003), and stronger than in the placebo condition (b = 0.07, F(480,1) = 4.54, p = p.034), indicating that PMFCA was stronger under levodopa as compared to placebo. Importantly, this interaction was not qualified by a triple nomination × informativeness × drug interaction (b = 0.02, t(480) = 0.33, p = 0.738), providing no evidence that the extent of PMFCA differed for informative and non-informative outcomes. No other effect pertaining to drug reached significance (*Appendix 1—table 3*).

To examine in more fine-grained detail whether an MFCA is indeed preferential, we calculated, for each participant, in each session (drug/placebo), and for each level of informativeness (informative/non-informative), the extent to which MFCA was preferential for the ghost-nominated as opposed to the ghost-rejected vehicle (as quantified by $c^{MF}_{nom,info} - c^{MF}_{rej,info}$, $c^{MF}_{nom,noninfo} - c^{MF}_{rej,noninfo}$; *Figure 5D*). Using a mixed effects model (see Materials and methods and *Appendix 1—table 3*), we regressed preferential MFCA (PMFCA), based on MB guidance on informativeness and drug session. We found a positive main effect for drug (b = 0.10, t(240) = 2.39, p = 0.017), but neither the main effect of informativeness (b = −0.03, t(240) = -0.57, p = 0.568) nor the informativeness × drug interaction (b = 0.02, t(240) = 0.34, p = 0.734) were significant. Using simple effects, MFCA preferred the ghost-nominated vehicle in the levodopa condition (b = 0.15, F(1,240) = 15.45, p < 0.001), while the same effect was only marginally significant in the placebo condition (b = 0.05, F(1,240) = 2.77, one-sided p = 0.048). No other effect pertaining to drug reached significance (*Appendix 1—table 3*). Thus, our computational modelling analysis indicates that PMFCA is boosted by levodopa as compared to placebo across informative and non-informative destinations.

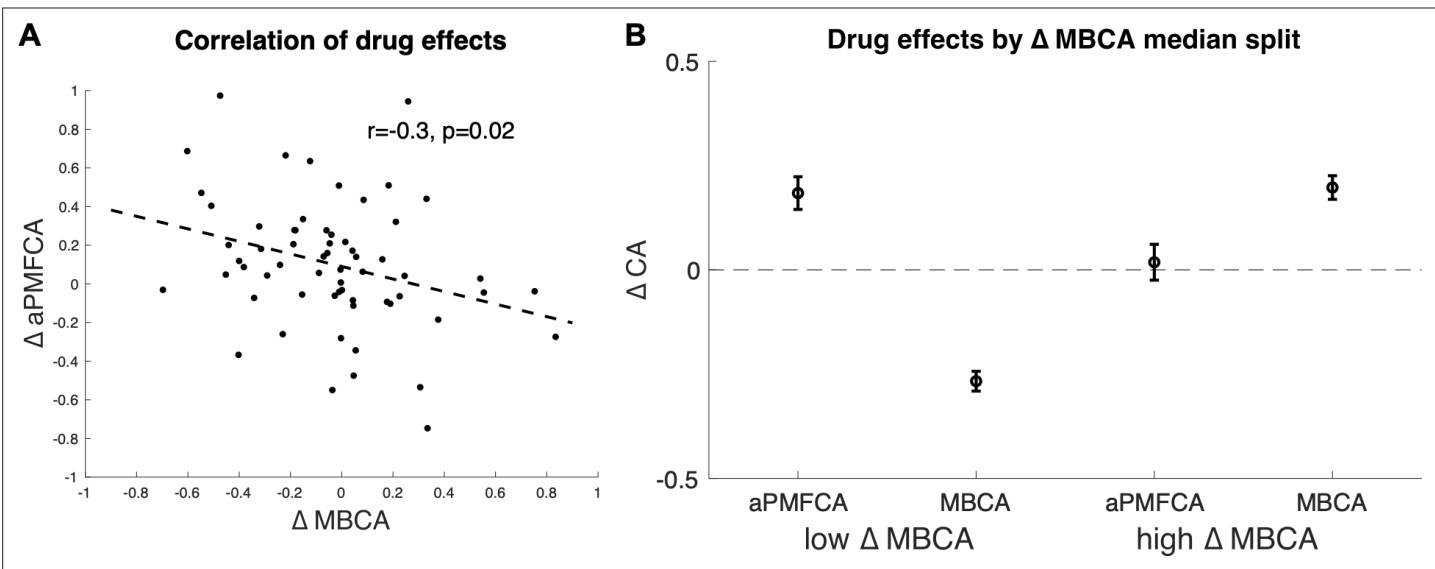

**Figure 6.** Inter-individual differences in drug effects in model-based credit assignment (MBCA) and in preferential model-based credit assignment (MFCA), averaged across informative and non-informative destinations (aPMFCA). (**A**) Scatter plot of the drug effects (levodopa minus placebo; ΔaPMFCA, ΔMBCA). Dashed regression line and Pearson r correlation coefficient (see *Appendix 1—figure 10* for an analysis that controls for parameter trade-off). (**B**) Drug effects in credit assignment (ΔCA) based on a median on ΔMBCA. Error bars correspond to SEM reflecting variability between participants. See *Appendix 1—figure 11*, for a report on inter-individual differences in drug effects related to working memory.

## Inter-individual differences in drug effects

Previous studies, using a task that cannot dissociate different MB processes, reported that boosting DA levels leads to enhanced MB choices (*Sharp et al., 2016* ; *Wunderlich et al., 2012*), an effect we did not observe at the group level on our measure of MBCA. To explore the possibility that drug effects for distinct MB processes (guidance of MFCA by MB inference vs. MBCA) are related, we analysed inter-individual differences in the impact of boosting DA levels on guidance of MFCA and on MBCA (for distribution of parameters, see *Appendix 1—table 4*). We first averaged PMFCA across informative and non-informative destinations (aPMFCA) as, in relation to a correlation between drug effects, we did not have specific assumptions regarding destination type. A correlation of drug-related changes (levodopa minus placebo; ΔaPMFCA and ΔMBCA) revealed a significant negative correlation (*Figure 6A*, Pearson r = −0.30, p = 0.017, Spearman r = −0.33, p = 0.009; see *Appendix 1—figure 10* for control analysis verifying that this negative correlation survives control for parameter trade-off).

We unpack this correlation further by next taking the median on ΔMBCA (–0.019) so as to split the sample into lower/higher median groups. The higher-median group showed a positive (M = 0.197, t(30) = 4.934, p < 0.001) and the lower-median group showed a negative (M = –0.267, t(30) = –7.97, p < 0.001), drug effect on MBCA, respectively (*Figure 6B*). In a mixed effects model (see Materials and methods), we regressed aPMFCA against drug and a group indicator of lower/higher median ΔMBCA groups. This revealed a significant drug × ΔMBCA-split interaction (b = −0.17, t(120) = −2.05, p = 0.042). In the negative ΔMBCA group (*Figure 6B*), a significantly positive drug effect on aPMFCA was detected (simple effect: b = 0.18, F(120,1) = 10.35, p = 0.002) while in the positive ΔMBCA group a drug-dependent change in aPMFCA was not significant (*Figure 6B*, simple effect: b = 0.02, F(120,1) = 0.10, p = 0.749).

We also report inter-individual differences in drug effects related to working memory in Appendix 1 (incl. *Appendix 1—figure 11*).

## Discussion

We show that enhancing DA boosted the guidance of MFCA by retrospective MB inference. Whereas both MF and MB influences were unaffected by the drug manipulation at the group level, analysis of inter-individual differences in drug effects showed that enhanced guidance of MFCA by retrospective MB inference was negatively correlated with drug-related change in pure MBCA. The findings provide, to our knowledge, the first human evidence that DA directly influences cooperative interactions between MB and MF systems, highlighting a novel role for DA in the guidance of MFCA by MB inference.

In our novel task, which allows us to separate pure MB control from a guidance of MFCA by MB inference, we found that, levodopa specifically enhanced the preferential MB guidance of MFCA in a situation where individuals could rely on retrospective MB inference to resolve state uncertainty. Thus, an MB instruction about *what* (unobserved or inferred) state the MF system might *learn about* was boosted under levodopa. In other words, DA boosts an exploitation of a model of task structure so as to facilitate retrospective MF learning about the past. These findings indicate an enhanced integration of MB information in DA signalling (*Sadacca et al., 2016*). Our results provide a fine-grained view of the DA dependency of at least two MB processes – with the specificities of our task allowing us to separate out on the group level a particular boosting of a guidance of MFCA by MB inference but not purely the extent of MB influence over control.

Using a task that was not designed to disentangle learning-based cooperation between MB and MF systems from MB influences per se (*Daw et al., 2011*), a positive relationship between DA and MB influences has been reported (*Deserno et al., 2015a*; *Doll et al., 2012*; *Sharp et al., 2016*; *Wunderlich et al., 2012*). These studies, as well as ours, remain equivocal regarding a potential striatal or prefrontal DA origin of these effects. An effect of levodopa on prefrontal DA levels can lead to the enhancement of general aspects of cognition, for example, working memory (*Cools and D'Esposito, 2011*), possibly depending on DA baseline in an inverted U-curved manner. Working memory is likely to be important for supporting the computationally sophisticated operation of an MB system (*Otto et al., 2013*). In this context, a primary drug effect on prefrontal DA might result in a boosting of purely MB influences. However, we found no such influence at a group level – unlike that seen previously in tasks that used only a single measure of MB influences (*Sharpe et al., 2017*; *Wunderlich*

*et al., 2012*). Our novel task systematically separates two MB processes: a guidance of MFCA by MB inference and pure MB control. While we found that only one of these, namely guidance of MFCA by MB inference, was sensitive to enhancement of DA levels at a group level, we did detected a negative correlation between the DA drug effects on MB guidance of MFCA and on pure MBCA. One explanation is that a DA-dependent enhancement in pure MB influences was masked by this boosting in guidance of MFCA, via an MB inference. In this regard, our data is suggestive of between-subject heterogeneity in the effects of boosting DA on distinct aspects of MB influences. This general suggestion is also in keeping with a trade-off between DA influences on these two MB aspects of behavioural control. In arbitrating between pure MB influences and retrospective MB inference to guide MFCA, participants may weigh respective cognitive costs vs. instrumental value. In independent recent work, a balance of costs and benefits was shown to be modulated by DA (*Westbrook et al., 2020*). In our study, MB guidance of MFCA was, at a group level, more sensitive to enhancing DA levels compared to pure MB influences. Hypothetically, this heightened sensitivity of a cooperative process intermediate to MF and MB systems could be a result of an efficient cost-benefit trade-off. For example, lower cognitive costs in MB guidance of MFCA, as compared to the costs of pure MB influences, may be pitted against benefits. Future studies should examine how the relative costs and benefits of planning vs. retrospective state inference are influenced by DA, which would also inform DA contributions to trade-offs pertaining to strategy selection.

A long-standing proposal that phasic DA relates to an MF learning signal predicts that the primary drug effects, reflecting an effect of levodopa on striatal DA levels, should be to speed up or bias MF learning (*Pessiglione et al., 2006*). With respect to our current task, and an established two-step task designed to dissociate MF and MB influences (*Daw et al., 2011*), there is no compelling human evidence for a direct impact of DA on MF learning or control (*Deserno et al., 2015a*; *Kroemer et al., 2019*; *Sharp et al., 2016*; *Wunderlich et al., 2012*; *Kroemer et al., 2019*). A commonality of our novel and the two-step task is the inclusion of dynamically changing reward contingencies. As MF learning is by definition incremental, slowly accumulating reward value over extended time periods, it follows that dynamic reward schedules may lessen a sensitivity to detect changes in MF processes (see *Doll et al., 2016* for discussion). In line with this, experiments in humans indicate that value-based choices performed without feedback-based learning (for reviews, see *Maia and Frank, 2011*; *Collins and Frank, 2014*), as well as learning in stable environments (*Pessiglione et al., 2006*), are susceptible to DA drug influences (or genetic proxies thereof) as expected under an MF RL account. Thus, the impact of DA boosting agents may vary as a function of contextual task demands. This resonates with features of our pharmacological manipulation using levodopa, which impacts primarily on presynaptic synthesis. Thus, instead of necessarily directly altering phasic DA release, levodopa impacts on baseline storage (*Kumakura and Cumming, 2009*), possibly reflected in overall DA tone. DA tone is proposed to encode average environmental reward rate (*Mohebi et al., 2019*; *Niv et al., 2007*), a putative environmental summary statistic that might in turn impact an arbitration between behavioural control strategies according to environmental demands (*Cools, 2019*).

While our study elucidates a DA influence on different MB processes, its scope means we are agnostic to potential underlying neural mechanisms. Animal and human work points to a role for orbitofrontal cortex (OFC) in representing a model of a task, including unobserved and inferred states (*Howard et al., 2020*; *Jones et al., 2012*; *Schuck et al., 2016*). Both MB processes under investigation in our task rely on access to the stable transition structure and may, thus, both depend similarly on this putative OFC function. Therefore, we do not consider it likely that an impact on OFC function per se accounts for DA modulation of pure MB influences as opposed to guidance of MFCA. Instead, we speculate that prefrontal working memory or hippocampal replay mechanisms are likely to contribute to these two MB processes.

During pure MB processing, the value of each visited destination is updated at a given trial n. Crucially, in trial n + 1, when two vehicles are offered for choice, values for the corresponding destination need to be retrieved to compute MB values for each vehicle. We assume this relies on working memory capacity whereby its implementation in dorso-lateral prefrontal function would be boosted via enhanced prefrontal DA levels. However, while working memory function could theoretically account for enhanced pure MB influences, MB choice planning could also be supported, at least partially, by enhancement of neural replay for sequences of destinations associated with the vehicles, where this is likely to be implemented at least in part within hippocampus. We note here that theoretical

treatments of hippocampal offline neural replay propose it informs credit assignment based on RPEs (*Mattar and Daw, 2018*).

We also speculate that guidance of MFCA by MB inference relies on a replay mechanism, where reverse replay could mediate a retrospective resolution of state uncertainty to enable an MFCA for the unobserved ghost-nominated action. A reverse replay has recently been shown to subserve non-local credit assignment based upon priority (*Liu et al., 2021a*). Under such an account, we would predict an enhanced reinstatement/replay of reward from non-informative destination and the inferred ghost's choice under the influence of levodopa. This could occur at the point in time when an informative destination is revealed, enabling MFCA to the ghost's choice based on rewards from both destinations. Whether this might occur indirectly, via an interaction with working memory that maintains a representation of the no longer available rewards from the non-informative destination, or as a direct consequence of the levodopa, is a question for future work. These potential mechanistic accounts are now testable using human magnetoencephalography data (*Liu et al., 2021b*). Indeed, under different experimental designs to our task, recent studies indicate that forward vs. backward replay play a key role in subserving different aspects of MF and MB control, including implementation of a non-local credit assignment (*Eldar et al., 2020*; *Liu et al., 2019*).

MB influences have previously been studied in relation to psychiatric symptom expression (*Deserno et al., 2015b*; *Gillan et al., 2016*; *Sebold et al., 2014*; *Sebold et al., 2017*; *Voon et al., 2015*). The enhancement of DA by drugs of abuse may boost MB processes in a manner that might serve to boost goal-directed drug seeking (*Hogarth, 2020*; *Simon and Daw, 2012*). Thus, instead of a drug-induced DA release merely reinforcing a set of drug-associated actions, an assumption in some models of addictive behaviours (*Dayan, 2009*; *Redish, 2004*), it might equally promote MB inference in a highly drug-specific manner. Our data suggests that cooperative processes at the intersection of MF and MB systems may be particularly sensitive to the impact of enhancing DA levels, and such processes might be prevalent at an early evolution of addictive behaviours when goal-directed drug seeking is more dominant (*Everitt and Robbins, 2005*). Moreover, emerging evidence for a positive influence of higher DA levels on MB influences, and as revealed in our study a differential sensitivity of distinct MB process to enhancing DA levels, provides a potential target to dissect variability in treatment responses to therapeutic agents which boost DA levels (e.g., psychostimulants in ADHD).

A limitation in our study is that the model-agnostic and computational modelling were not entirely consistent in two aspects. Firstly, in contrast to *Moran et al., 2019*, the model-agnostic analysis of repeat vs. switch trials, providing a measure of preferential guidance of MF choice repetition, was not significant under placebo alone. The corresponding effect reached (one-sided) significance based on the parameters of the computational model. Secondly, we did not find a drug enhancing effect on guidance of MF choice repetition based on reward at the non-informative destinations, as assessed in the condition of clash trials. However, when analysing the parameters of the computational model, there was a significant drug effect indicating enhanced MFCA independent of the informativeness of the destination (not qualified by a triple interaction). We suggest that these types of discrepancies between studies are to be expected when effects of interest are weak to moderate.

Computational modelling may have superior sensitivity when capturing subtle effects of interest, particularly as it exploits influences (decaying) from an entire past history of choice and outcomes, and not only from the preceding trial as in model-agnostic analyses. Further, the model-agnostic approach examines distinct subsets of trial-transition for the different 'signatures' (e.g., each of the 'repeat', 'switch', and 'clash' conditions are informed by 1/9 of all trial transitions), whereas computational modelling parameters are informed by all trials in the task, rendering this approach potentially more powerful. Thus, whereas a model-agnostic approach does not allow for a direct comparison of PMFCA for the informative and non-informative destinations (since preferentiality effects are measured in different ways, which are not directly comparable, for these two destination types), a computational-modelling approach quantifies informative/non-informative effects within a common scale, allowing comparison and aggregation of PMFCA effects across outcome types. The computational models fitted to choice data are conceptually very similar to the those reported in *Moran et al., 2019*, though we changed the models' parameterizations (as detailed in Appendix 1). Lastly, we again acknowledge that the effects reported were small to moderate, particular with respect to inter-individual differences in drug effects.

In sum, our study provides evidence that DA enhances cooperative interactions intermediate to MB and MF systems. The findings provide a unified perspective on previous research in humans and animals, suggesting a closely integrated architecture in relation to how MF and MB systems interact under the guidance of DA-mediated cooperation so as to improve learning. In future work, we aim to examine the precise neural mechanisms for this DA-mediated cooperation between MB and MF system and this may inform the involvement of MB control in the development of impulsive and compulsive psychiatric symptoms.

## Materials and methods

### Procedures

The study was approved by the University College London Research Ethics Committee (Project ID 11285/001). Subjects gave written informed consent before the experiment. Previous studies on DA drug effects on MB control were conducted in rather small sample size (*Sharp et al., 2016*; *Wunderlich et al., 2012*), which are prone to overestimate effect sizes (*Button et al., 2013*). Thus, we assumed small to moderate effect size (d = 0.35) to determine sample size, which is also in line with a more recent DA drug study on MB control (*Kroemer et al., 2019*). A total of 64 participants (32 females) completed a bandit at each of the two sessions with drug or placebo in counterbalanced order in a double-blinded design. One participant failed to reach required performance during training (see below) and task data could not be collected. Out of remaining 63 participants, one participant experienced side effects during task performance and was therefore excluded. Results reported above are based on a sample of n = 62. All participants attended on two sessions approximately 1 week apart. Participants were screened to have no psychiatric or somatic condition, no regular intake of medication before invitation and received a short on-site medical screening at the beginning of their day 1 visit. At the beginning of the day 2 visit, they performed a working memory test, the digit span, which was thus only collected once.

### Drug protocol

The order of drug and placebo was counterbalanced. The protocol contained two decision-making tasks, which started at least 60 min after ingestion of either levodopa (150 mg of levodopa +37.5 mg of benserazide dispersed in orange squash) or placebo (orange squash alone with ascorbic acid). Benserazide reduces peripheral metabolism of levodopa, thus, leads to higher levels of DA in the brain and minimizes side effects such as nausea and vomiting. To achieve comparable drug absorption across individuals, subjects were instructed not to eat for up to 2 hr before commencing the study. Repeated physiological measurements (blood pressure and heart rate) and subjective mood rating scales were recorded under placebo and levodopa. A doctor prepared the orange squash such that data collection was double-blinded.

### Task description

Participants were introduced to a minor variant of a task developed by *Moran et al., 2019*, using pictures of vehicles and destinations rather than objects and coloured rooms, and lasting slightly less time. Based on previous data collection with this task (*Moran et al., 2019*), we reasoned that a less arbitrary connection and a natural meaning (vehicles and destination as compared of objects and colours) would make the cover story more plausible and entertaining and, thus, could facilitate compliance in our pharmacological within-subjects study. Further, for vehicle-destination mappings there is a broad range of stimuli available for two version of the task. The only further change was that the task was slightly shortened from 504 (7 blocks with 72 trials) to 360 (5 blocks with 72) trials per session (720 trials across drug and placebo trials). This latter decision was made based on the data by Moran et al., which indicated comparable results when cutting down to 360 trials and was preferable for the pharmacological design. No other changes were implemented to the task.

The task was presented as a treasure hunt called the 'Magic Castle'. Before playing the main task, all participants were instructed that they can choose out of four vehicles from the Magic Castle's garage that each vehicle could take them to two destinations (see *Figure 1B*). The mapping between vehicles and destination was randomly created for each participant and each session (sessions also had different sets of stimuli) but remained fixed for one session. They were then extensively trained

on the specific vehicle-destination mapping. In this training, participants first saw a vehicle and had to press the space bar in self-paced time to subsequently visit the two associated destinations in random order. There were 12 repetitions per vehicle-destination mapping (48 trials). Following this initial training, participants responded to two types of quiz trials where they were either asked to match one destination out of two to a vehicle, or to match one vehicle out of two to a destination within a time limit of 3 s (eight trials per quiz type). To ensure each and every participant had the same level of knowledge of the transition structure, each quiz trial had to be answered correctly and in time (<3 s) within the placebo and drug sessions (which had different sets of stimuli). Otherwise, a further training session followed but now with only four repetitions per vehicle-destination mapping (16 trials), followed again by the two types of quiz trials. This was repeated until criterion was reached. The criterion was identical for each participant in placebo and drug sessions, while the number of training cycles until criterion was reached could vary. The average number of training cycles (drug: mean 3.2381, std 3.0518, min 1, max 18; placebo: mean 3.6774, std 2.7328, min 1, max 13) did not differ between sessions (Wilcoxon signed rank test, p = 0.1942).

Participants were then introduced to the general structure of standard trials of the bandit task (18 practice trials). This was followed by instructions introducing the ghost trials, which were complemented by another 16 practice trials including standard and ghost trials. Before starting the actual main experiment, subjects were required to perform a short refresher training of the vehicle-destination mappings (with four repetitions per vehicle-destination mapping), followed by a requirement to pass the same quiz trials as described above. If they failed to pass at this stage, the refresher training was repeated with two repetitions per vehicle-destination mapping until such time as the quiz was passed. The average cycles of refresher training (drug: mean 2.2222, std 1.5600, min 1, max 10; placebo: mean 2.0476, std 1.4528, min 1, max 9) did not differ between sessions based on a Wilcoxon signed rank test (p = 0.3350).

During the subsequent main task, participants should try to maximize their earnings. In each trial, they could probabilistically find a treasure (reward) at each of the two destinations (worth one penny). Reward probabilities varied over time independently for each of the four destinations according to Gaussian random walks with boundaries at p = 0 and p = 1 and a standard deviation of 0.025 per trial (*Figure 1C*). Random walks were generated anew per participant and session. A total of 360 trials split in 5 blocks of each 72 trials were played with short enforced breaks between blocks. Two of three trials were 'standard trials', in which a random pair of objects was offered for choice sharing one common outcome (choice time ≤ 2 s). After making a choice, they visited each destination subsequently in random order. Each destination was presented for 1 s and overlaid with treasure or not (indicating a reward or not). The lag between the logged choice and the first destination as well as between first and second destinations was 500 ms. Every third trial was an 'uncertainty trial' in which two disjoint pairs of vehicles were offered for choice. Crucially, each of the presented pairs of vehicles shared one common destination. Participants were told before the main task that after their choice of a pair of vehicles, the ghost of the Magic Castle would randomly pick one vehicle out of the chosen pair. Because this ghost was transparent, participants could not see the ghost's choice. However, participants visited the two destinations subsequently and collected treasure reward (or not). Essentially, when the ghost nominated a vehicle, the common destination was presented first and the destination unique to this vehicle was presented second. At this time of presentation of the unique destination, participants could retrospectively infer the choice made by the ghost. In standard trials n + 1 following uncertainty trials n, we separate three conditions depending on the presented vehicle pair. A repeat trial presented the ghost-nominated object alongside its vertical counterpart from the previously non-selected pair, a switch trial presented the ghost-rejected object alongside its vertical counterpart from the previously non-selected pair and a clash trial presented the previously selected pair. Trial timing was identical for standard and ghost trials.

Each session (drug as well as placebo) had 360 trials, resulting in 720 trials in total. One-third of trials (120 per session) were standard trials (following standard trials) and included 30 presentations of each of four eligible pairs of vehicles in random order. Half of these trials are used for each of the analyses of MF and MB signatures (*Figure 2*). This resulted in a maximum of 60 trials for the analysis of each of the MF and MB signatures. For each session, every third trial was an uncertainty trial (120 trials), which were always followed by a standard trial, where a third of these trials (40 trials) contributed to each of the 'repeat', 'switch', and 'clash' conditions (*Figure 4*). Accounting for missed choices,

this resulted in the following trial averages per condition and per session (placebo/drug): MF-placebo 58.07, MF-drug 59.15; MB-placebo 58.00, MB-drug 57.01; repeat-placebo 39.42, repeat-drug 39.31; switch-placebo 39.34, switch-drug 39.10; clash-placebo 39.34, clash-drug 39.32.

## Model-agnostic analysis

Model-agnostic analyses were performed with generalized mixed effects models using MATLAB's 'fitglme' function with participants (PART) serving as random effects with a free covariance matrix (which was restricted in case of non-convergence). All models included the variable ORDER as regressor (coded as +0.5 for the first and –0.5 for the second session) to control for unspecific effects.

The analysis of MF and MB contributions is restricted to standard trials followed by a standard trial. For MF contributions, we consider only a trial n+1, which offers the trial n chosen object for choice (against another object). Regressors C (common destination) and U (unique destination) indicated whether rewards were received at trial n (coded as +0.5 for reward and –0.5 for no reward) and were included to predict the variable REPEAT indicating whether the previously chosen vehicle was repeated or not. The variable DRUG was included as regressor indicating within-subject levodopa or placebo session (coded as +0.5 for levodopa and –0.5 for placebo). The binomial model, in Wilkinson notation, can be found in *Appendix 1—table 1*. For MB contributions, we specifically examined trials in which the trial n chosen vehicle was excluded on trial n + 1. The regressors C, PART, and DRUG were coded as for the analysis of the MF contribution. One additional regressor P was included, which coded the continuous reward probability of the common destination and was centred by subtracting 0.5. These regressors were included to predict the variable GENERALIZE indicated whether the choice on trial n + 1 was generalized (choosing the vehicle not shown in trial n + 1 that shares a destination with the trial n chosen vehicle). The logistic model, in Wilkinson notation, and detailed statistics can be found in *Appendix 1—table 1*. Note that because this model included a continuous regressor, it was a logistic model (a special case of the binomial model when all predictor-combinations occur once).

The analysis of how retrospective MB inference preferentially guides MFCA focused on standard (repeat and switch) trials n + 1 following uncertainty trials n. For repeat trials, we computed the proportion of choice repetitions (choosing the previously ghost-nominated vehicle) after a reward minus no-reward at the informative destination (reflecting the main effect of the informative destination's reward, 'I'). Similarly, for switch trials we computed the proportion of choice-switches (choosing the previously ghost-rejected vehicle) after a reward minus no-reward at the informative destination. These reward effects were averaged across instances where the non-informative vehicle was previously rewarded or unrewarded (thereby controlling for influences from this reward). Reward effects were then subjected to a linear mixed effects model as dependent variable with nomination (NOM; nominated/rejected coded as +0.5 and –0.5) and, as before, DRUG and PART as predictors. The model, in Wilkinson notation, and detailed statistics can be found in *Appendix 1—table 2*. An alternative detailed analysis using a binomial mixed effects model, which focuses on choice proportions as a function of whether the informative destination was previously rewarded or not (in contrast to reward effects), is reported in Appendix 1.

Another model-agnostic analysis using a binomial mixed effects model examined learning for the ghost-nominated and -rejected vehicles based on the uncertainty trial n non-informative destination and therefore focused on n + 1 'clash' trials, which offer for choice the same pair of objects as chosen on the previous uncertainty trial (the ghost-nominated and ghost-rejected objects). Choice repetition was defined as choice of the ghost-nominated vehicle from uncertainty trial n indicated by the variable REPEAT. Regressors N (non-informative destination) and I (informative destination) indicated whether rewards were received at trial n (coded as +0.5 for reward and –0.5 for no reward). Regressors PART and DRUG are coded as previously. The model, in Wilkinson notation, and detailed statistics can be found in *Appendix 1—table 2*.

## Computational models

We formulated a hybrid RL model to account for the series of choices for each participant. In the model, choices are contributed by both the MB and MF systems. The MF system caches a $Q^{MF}$ value for each vehicle, subsequently retrieved when the vehicle is offered for choice. During learning on standard trials, following reward feedback, rewards from the two visited destinations are used to update the $Q^{MF}$ value for the chosen vehicle as follows:

$$Q^{MF}\ (chosenvehicle)$$
$$\leftarrow \left(1 - f^{MF}\right) * Q^{MF}\left(chosenvehicle\right) + c^{MF}_{standard} \qquad (2)$$
$$*(r_1 + r_2)$$

where $c^{MF}_{standard}$ is a free MFCA parameter on standard trials and the r's are the rewards for each of the two obtained outcomes (coded as one for reward or –1 for non-reward) and $f^{MF}$ (between 0 and 1) is a free parameter corresponding to forgetting in the MF system.

During learning on uncertainty trials, the MF values of the ghost-nominated and ghost-rejected options were updated according to:

$$Q^{MF}\ (nominatedvehicle)$$
$$\leftarrow \left(1 - f^{MF}\right) * Q^{MF}\left(nominatedvehicle\right) + c^{MF}_{nom,info} * r_{info} + c^{MF}_{nom,noninfo} \qquad (3)$$
$$*r_{noninfo}$$

$$Q^{MF}\ (rejectedvehicle)$$
$$\leftarrow \left(1 - f^{MF}\right) * Q^{MF}\left(rejectedvehicle\right) + c^{MF}_{rej,info} * r_{info} + c^{MF}_{rej,noninfo} \qquad (4)$$
$$*r_{noninfo}$$

where the c's are free MFCA parameters on uncertainty trials for each destination (informative/non-informative) and vehicle type (ghost-nominated/rejected) in the chosen pair. The r's are rewards (once more, coded as 1 or –1) for the informative and non-informative outcomes.

The MF values of the remaining vehicles (three on standard trials; two on uncertainty trials) were subject to forgetting:

$$Q^{MF}\left(nonchosenvehicles\right) \leftarrow \left(1 - f^{MF}\right) * Q^{MF}\left(nonchosenvehicles\right) \qquad (5)$$

Unlike MF, the MB system maintains $Q^{MB}$ values for the four different destinations. During choices the $Q^{MB}$ value for each offered vehicle is calculated based on the transition structure (i.e., the two destinations associated with a vehicle):

$$Q^{MB}\left(vehicle\right) = Q^{MB}\left(destination1\right) + Q^{MB}\left(destination2\right) \qquad (6)$$

Following a choice (on both standard and uncertainty trials), the MB system updates the $Q^{MB}$ values of each of the two observed destination based on its own reward:

$$Q^{MB}\left(destination\right) \leftarrow \left(1 - f^{MB}\right) * Q^{MB}\left(destination\right) + c^{MB} * r \qquad (7)$$

where $f^{MB}$ (between 0 and 1) is a free parameter corresponding to forgetting in the MB system, $c^{MB}$ is a free MBCA parameter and r corresponds to the reward (1 or –1) obtained at the destination.

Our model additionally included progressive perseveration for vehicles. After each standard trial the perseveration values of each of the four vehicles updated according to:

$$PERS\left(vehicle\right) \leftarrow \left(1 - f^P\right) * PERS\left(vehicle\right) + pr_{standard} * 1_{vehicle=chosen} \qquad (8)$$

where $1_{vehicle=chosen}$ is the chosen vehicle indicator, $pr_{standard}$ is a free perseveration parameter for standard trials, and $f^P$ (between 0 and 1) is a free perseveration forgetting parameter. Similarly after each uncertainty trials, perseverations values were updated according to:

$$PERS\left(vehicle\right) \leftarrow \left(1 - f^P\right) * PERS\left(vehicle\right) + pr_{uncertainty} * 1_{vehicle=nom} \qquad (9)$$

where $1_{vehicle=nom}$ is the ghost-nominated vehicle indicator, and $pr_{uncertainty}$ is a free perseveration parameter for uncertainty trials.

During a standard trial choice a net Q value was calculated for each offered vehicle:

$$Q_{net}\left(vehicle\right) = Q^{MB}\left(vehicle\right) + Q^{MF}\left(vehicle\right) + PERS\left(vehicle\right) \qquad (10)$$

Similarly, during an uncertainty-trial choice the $Q_{net}$ value of each offered vehicle pair was calculated as a sum of the MB, MF, and PERS values of that pair. MF, MB, and PERS values for a vehicle pair in turn were each calculated as the corresponding average value of the two vehicles in that pair. For example:

$$Q^{MF}\left(vehiclepair\right) \leftarrow \frac{Q^{MF}\left(vehicle1\right)+Q^{MF}\left(vehicle2\right)}{2} \tag{11}$$

The $Q_{net}$ values for the two vehicles offered for choice on standard trials are then injected into a soft-max choice rule such that the probability to choose an option is:

$$Prob\left(vehicle\right) = \frac{e^{Q_{net}\left(vehicle\right)}}{e^{[Q_{net}(vehicle)+Q_{net}(other\ vehicle)]}} \tag{12}$$

Similarly, on uncertainty trials the probability to choice a vehicle pair was based on soft-maxing the net Q values of the two offered pairs. $Q^{MF}$ and $PERS$ person-values and $Q^{MB}$ values were initialized to 0 at the beginning of the experiment.

## Model comparison and fitting

Our full hybrid agents, which allowed for contributions from both an MB and an MF system, served as a super-model in a family of six nested sub-models of interest: (1) a pure MB model, which was obtained by setting the contribution of the MF to 0 (i.e., $c^{MF}_{standard} = c^{MF}_{nom,info} = c^{MF}_{nom,noninfo} = c^{MF}_{rej,info} = c^{MF}_{rej,noninfo} = 0$), (2) a pure MF-action model, which was obtained by setting the contribution of the MB system to choices to 0 (i.e., $c^{MB} = 0$; note that in this model, MB inference was still allowed to guide MF inference), (3) a 'no informativeness effect on MFCA' sub-model obtained by constraining equality between the MFCA for the informative and non-informative destination (i.e., $c^{MF}_{nom,info} = c^{MF}_{nom,noninfo}$ , $c^{MF}_{rej,info} = c^{MF}_{rej,noninfo}$) , (4) a 'no MB guided MFCA' sub-model obtained by constraining equality between the MFCA parameters, for both the informative and non-informative destination, for the ghost-nominated and -rejected objects ($c^{MF}_{nom,info} = c^{MF}_{rej,info}$ , $c^{MF}_{nom,noninfo} = c^{MF}_{rej,noninfo}$), (5) a 'no MB guidance of MFCA for the informative outcome' obtained by constraining equality between the MFCA parameters for the ghost-nominated and -rejected objects for the informative outcome ($c^{MF}_{nom,info} = c^{MF}_{rej,info}$ ), and (6) a 'no MB guidance of MFCA for the non-informative outcome' which was similar to five but for the non-informative outcome ($c^{MF}_{nom,noninfo} = c^{MF}_{rej,noninfo}$ ).

We conducted a bootstrapped generalized likelihood ratio test, BGLRT (**Moran and Goshen-Gottstein, 2015**), for the super-model vs. each of the sub-models separately. In a nutshell, this method is based on the classical-statistics hypothesis testing approach and specifically on the generalized-likelihood ratio test (GLRT). However, whereas GLRT assumes asymptotic chi-squared null distribution for the log-likelihood improvement of a super-model over a sub-model, in BGLRT these distributions are derived empirically based on a parametric bootstrap method. In each of our model comparison the sub-model serves as the H0 null hypothesis whereas the full model as the alternative H1 hypothesis. For each participant and drug condition, we created 1001 synthetic experimental sessions by simulating the sub-agent with the ML parameters on novel trial sequences which were generated as in the actual data. We next fitted both the super-agent and the sub-agent to each synthetic dataset and calculated the improvement in twice the logarithm of the likelihood for the full model. For each participant and drug condition, these 1001 likelihood-improvement values served as a null distribution to reject the sub-model. The p-value for each participant in each drug condition was calculated based on the proportion of synthetic dataset for which the twice logarithm of the likelihood improvement was at least as large as the empirical improvement. Additionally, we performed the model comparison at the group level. We repeated the following 10,000 times. For each participant and drug condition we chose randomly, and uniformly, one of his/her 1000 synthetic twice log-likelihood super-model improvements and we summed across participant and drug conditions. These 10,000 obtained values constitute the distribution of group super-model likelihood improvement under the null hypothesis that a sub-model imposes. We then calculated the p-value for rejecting the sub-agent at the group level as the proportion of synthetic datasets for which the super-agent twice logarithm of the likelihood improvement was larger or equal to the empirical improvement in super-model, summed across participants. Results, as displayed in **Appendix 1—figure 7** in detail, fully supported the use of our full model including all effects of interest regarding MFCA in uncertainty trials. This was replicated when using our current novel model on the data from **Moran et al., 2019** (**Appendix 1—figure 8**).

We next fit our choice models to the data of each individual, separately for each drug condition (levodopa/placebo) maximizing the likelihood (ML) of their choices (we optimized likelihood using MATLAB's 'fmincon', with 200 random starting points per participant * drug condition; *Appendix 1—table 4* for distribution best-fitting parameters).

## Model simulations

To generate model predictions with respect to choices, we simulated for each participant and each drug condition, 25 synthetic experimental sessions (novel trial sequences were generated as in the actual experiment), based on ML parameters obtained from the corresponding model fits. We then analysed these data in the same way as the original empirical data (but with datasets that were 25 times larger, as compared to the empirical data, per participant). Results are reported in *Appendix 1— figures 1 and 3*. We also verified recoverability of the credit-assignment model parameters was good (see *Appendix 1—figure 9*).

## Analysis of model parameters

All models included the variable ORDER as regressor (coded as +0.5 for the first and –0.5 for the second session) to control for unspecific effects and participants (PART) served as random effects. Details are reported in *Appendix 1—table 3*. For each participant in each drug condition, we obtained, based on the full model, four MFCA parameter estimates corresponding to destination (informative/non-informative) and vehicle (nominated/rejected) types. We conducted a mixed effects linear model (again implemented with MATLAB's function 'fitglme') with TYPE (nominated/rejected coded as +0.5 and –0.5), INFO (informative/non-informative coded as +0.5 and –0.5) and DRUG (drug/placebo coded as +0.5 and –0.5) as regressors. The model, in Wilkinson notation, can be found in *Appendix 1—table 3*. After finding significant drug by NOM * DRUG interaction, we followed this up in detail: we calculated for each participant in each drug condition and for each destination type the 'preferential MFCA' (denoted PMFCA) effect as the difference between the corresponding nominated and rejected MFCA parameters. We next ran a mixed effects model for PMFCA. Our regressors were the destination type (denoted INFO; coded as before) and DRUG (coded as before). The model, in Wilkinson notation, can be found in *Appendix 1—table 3*.

## Relationship between drug effects

To explore the relation between drug effects on guidance of MFCA by MB inference and on MBCA, we first averaged PMFCA (as described) across informative and non-informative destination (aPMFCA) and, second, tested a Pearson correlation between drug effects (levodopa minus placebo; ΔaPMFCA and ΔMBCA). To further unpack this correlation, we performed a median split on ΔMBCA. In a mixed effects model, we regressed average ΔaPMFCA against drug and a group indicator of whether ΔMBCA was higher (coded as +0.5) or lower (–0.5) as compared to the median.

## Acknowledgements

RJD is supported by a Wellcome Trust Investigator Award (098362/Z/12/Z) under which the above study was carried out. This work was carried out whilst RJD was in receipt of a Lundbeck Visiting Professorship (R290-2018-2804) to the Danish Research Centre for Magnetic Resonance. RM is supported by the Max Planck Society as was LD at the time the study was performed. The UCL-Max Planck Centre for Computational Psychiatry and Ageing is funded by a joint initiative between UCL and the Max Planck Society. RJD and LD are supported by a grant from the German Research Foundation (DFG TRR 265, 402170461, project A02), as was YL at the time the study was performed. PD is funded by the Max Planck Society and the Humboldt Foundation. We thank Sam Wray for support with data collection. For the purpose of Open Access, the authors have applied a CC BY public copyright license to any Author Accepted Manuscript version arising from this submission.

## Additional information

### Funding

| Funder | Grant reference number | Author |
|---|---|---|
| Wellcome Trust | 098362/Z/12/Z | Raymond J Dolan |
| Max-Planck-Gesellschaft | Open-access funding | Lorenz Deserno<br>Rani Moran<br>Peter Dayan<br>Raymond J Dolan |
| Deutsche Forschungsgemeinschaft | 402170461 | Lorenz Deserno<br>Raymond J Dolan |

The funders had no role in study design, data collection and interpretation, or the decision to submit the work for publication.

### Author contributions

Lorenz Deserno, Conceptualization, Data curation, Formal analysis, Project administration, Resources, Software, Writing – original draft, Writing – review and editing; Rani Moran, Conceptualization, Formal analysis, Methodology, Resources, Software, Writing – original draft, Writing – review and editing; Jochen Michely, Project administration, Writing – review and editing; Ying Lee, Data curation, Project administration, Writing – review and editing; Peter Dayan, Supervision, Writing – review and editing; Raymond J Dolan, Funding acquisition, Supervision, Writing – review and editing

### Author ORCIDs

Lorenz Deserno http://orcid.org/0000-0001-7392-5280
Rani Moran http://orcid.org/0000-0002-7641-2402
Jochen Michely http://orcid.org/0000-0003-3072-2330
Ying Lee http://orcid.org/0000-0001-9491-4919
Peter Dayan http://orcid.org/0000-0003-3476-1839
Raymond J Dolan http://orcid.org/0000-0001-9356-761X

### Ethics

Human subjects: The study was approved by the University College London Research Ethics Committee (Project ID 11285/001). Subjects gave written informed consent before the experiment.

### Decision letter and Author response

Decision letter https://doi.org/10.7554/eLife.67778.sa1
Author response https://doi.org/10.7554/eLife.67778.sa2

## Additional files

### Supplementary files

• Transparent reporting form

### Data availability

Necessary source data files are openly available at: https://osf.io/4dfkv/.

The following dataset was generated:

| Author(s) | Year | Dataset title | Dataset URL | Database and Identifier |
|---|---|---|---|---|
| Deserno L | 2021 | Dopamine enhances model-free credit assignment through boosting of retrospective model-based inference | https://osf.io/4dfkv/ | Open Science Framework, 4dfkv |

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

# Appendix 1

## Repeat and switch standard trials following uncertainty trials

We showed previously on 'repeat' trials (*Moran et al., 2019*), a positive effect of an informative destination reward (on trial n) on choice-repetition implicates MFCA to the ghost-nominated object (while the MB system knows that the value of the informative destination favours both vehicles on trial n + 1). We also ran a separate analysis that examined MFCA for the ghost-nominated alone. In trial n + 1 'repeat' trials, the ghost-nominated vehicle from trial n is offered for choice alongside a vehicle from the trial n non-chosen pair that shares the inference-allowing destination with the ghost-nominated object. Choice repetition was defined as choice of the ghost-nominated vehicle from uncertainty trial n as indicated by the variable REPEAT. Regressor PART is coded as previously. Regressors N (non-informative destination) and I (informative destination) indicate whether a reward was received at the destinations or not in trial n (coded as +0.5/–0.5). The model is REPEAT ~ N*I + (N*I | PART). This showed a main effect for the informative (I) destination (b = 0.60, t(492) = 7.90, p = 1e-14), supporting MFCA to the ghost-nominated object. Additionally, we found a main effect for the non-informative (N) destination (b = 1.22, t(492) = 13.49, p = 1e-35) as predicted by both MF and by MB contributions, and a significant interaction between the informative and non-informative destinations (b = 0.34, t(492) = 2.29, p = 0.02). See *Appendix 1—figure 5 and A & B*.

We showed previously on switch trials that a positive main effect of the informative outcome reward on choice-switching implicates MFCA for the ghost-rejected vehicle (because the MB system knows the informative destination is unrelated to both vehicles on trial n + 1). A second separate analysis examined MFCA for the ghost-rejected vehicle. In uncertainty trial n +1 'switch' trials, the ghost-rejected vehicle from trial n is offered for choice alongside a vehicle from the trial n non-chosen pair that shares a destination with the ghost-rejected object. Choice switching was defined as choice of the ghost-rejected vehicle from uncertainty trial n as indicated by the variable SWITCH. Regressors PART, N, and I are coded as previously. The model is SWITCH ~ N*I + (N*I | PART). This showed a main effect for the reward at informative destination (b = 0.38, t(491) = 5.82, p = 1e-8), supporting MFCA to the ghost-rejected vehicle. While this challenges any notion of *perfect* MB guidance of MFCA, it is consistent with the possibility that some participants, at least some of the time, do not rely on MB inference because when MB inference does not occur, or when it fails to guide MF credit assignment, the MF system has no basis to assign credit unequally to both vehicles in the selected pair. Additionally, we found a main effect for the non-informative destination reward (b = 0.95, t(491) = 12.61, p = 9e-32), as predicted by an MF credit assignment to the ghost-rejected vehicle account but also by MB contributions. We found no significant interaction between rewards at the informative and non-informative destinations(b=-0.20, t(492)=-1.45, p=0.15). In *Appendix 1—figure 5*, we plot empirical choice proportions from both repeat and switch conditions (reflecting the effects reported above) in the manner as in the original paper by *Moran et al., 2019* but separately for drug and placebo conditions. See *Appendix 1—figure 5C & D*.

## Analyses of drug effects combined across repeat and switch standard trials following uncertainty trials

Our analysis of repeat/switch trials as reported in the main manuscript relies on a linear mixed effects model of reward effects as described in the Materials and methods. The reason to perform the analysis this way was to keep the model simple, which is in line with analysis approach reported in *Moran et al., 2019*. A binomial mixed effects model including data from repeat and switch trials would include additional two regressors, reward/no-reward at the informative destination and reward/no-reward at the non-informative destination, leading to a total of five regressors. This results in a complex five-way model. Upon request, we also ran this complex binomial mixed effects models. In this model, we regressed proportion of choices (rather than reward effects) of a vehicle from the previously chosen vehicle pair on predictors of interest including reward at the informative destination (I), nomination trial type (NOM), and drug but also rewards from the non-informative destination (N) and order as control (this yields five predictors in total each with two levels coded as +0.5 and –0.5). Specifically, we formulate the model as:

$$\text{CHOICE} \sim 1 + (\text{I*NOM*DRUG} + \text{N})*\text{ORDER} + (\text{I*NOM*DRUG} + \text{N} + \text{ORDER|PART})$$

Importantly, we find a positive significant three-way informative-reward × nomination × drug interaction effect (b = 0.39, t(973) = 2.00, p = 0.046). This interaction was not qualified by any higher order interactions. To interpret this interaction, we examined the simple two-way informative-reward × trial-type interactions for levodopa and placebo. For levodopa, we found a positive two-way interaction (b = 0.42, F(1,973) = 9.872, p = 0.002), which was explained by the fact that the simple informative reward effect for repeat trials (b = 0.68, F(1,973) = 46.06, p = 2e-11) was stronger than for switch trials (b = 0.27, F(1,973) = 8.46, p = 0 .004). Thus, for levodopa, we found PMFCA. In contrast, for placebo, the two-way informative-reward × nomination interaction was non-significant (b = 0.03, F(1,973) = 0.031, p = 0.859), providing no evidence for PMFCA.

## Absence of drug effects on perseveration and forgetting parameters of the computational model

No difference between drug conditions was observed for perseveration parameter on standard trials (t(61) = 0.48, p = 0.63), perseveration parameter on uncertainty trials (t(61) = 0.51, p = 0.61), MF forgetting parameter (t(61) = 1.37, p = 0.17), MB forgetting parameter (t(61) = –0.33, p = 0.74), perseveration forgetting parameter (t(61) = 0.30, p = 0.77).

## Drug effect on guidance of MFCA and working memory

For each participant and destination type (informative/non-informative), we contrasted the 'PMFCA' estimates (as defined in the main results section) for levodopa minus placebo to obtain a drug-induced PMFCA effect. For each destination, we calculated across-participants Spearman correlations between these drug-induced effects and working memory (WM) as ascertained with the digit span test. We compared the two correlations (for informative and non-informative destinations) using a permutation test. First, we z-scored the PMFCA separately for each destination type. Next we repeated the following steps (1–3), 10,000 times: (1) For each participant we randomly reshuffled (independent of other participants) the outcome type labels 'informative' and 'non-informative', (2) we calculated the 'synthetic' Spearman correlations between drug-induced PMFCA effects and WM for each outcome type subject to the relabelling scheme, and (3) we subtracted the two correlations (non-informative minus informative). These 10,000 correlation differences constituted a null distribution for testing the null hypothesis that the two correlations are equal. Finally, we calculated the p-value for testing the hypothesis of a stronger correlation for the non-informative destination as the percentage (of the 10,000) synthetic correlation differences that were at least as large (in absolute value) as the empirical correlation difference. Subjects' WM capacity showed a positive across-participants Spearman correlation with the drug effect (levodopa vs. placebo) on PMFCA in the non-informative (r = 0.278, p = 0.029, *Appendix 1—figure 11*), but not for the informative destination (r = –0.057, p = 0.659, *Appendix 1—figure 11*). The difference between these correlations was significant (p = 0.044, permutation test; see Materials and methods).

## Drug effect on MBCA and WM

No moderating effect of WM on a drug-dependent difference in MBCA was observed (r = −0.07, p = 0.59). WM correlated positively (but non-significant) with MBCA separately in each session (placebo: r = 0.21, p = 0.08; drug: r = 0.15, p = 0.23).

## Clarifications regarding the computational modelling

With respect to the computational models fitted to the choice data, while the reported models are conceptually similar to *Moran et al., 2019*, we changed the model's and sub-model's parameterization. This was motivated by the following consideration. In this previous paper we found consistent preferential-MFCA model-agnostic effects across both the informative and non-informative destinations. Therefore, our models used separate Rescorla-Wagner updates for each bandit (the ghost-nominated and the ghost-rejected) based on the total rewards from the informative and non-informative outcomes. PMFCA (based on MB guidance) then manifested as a higher learning rate for the ghost-nominated as compared to the ghost-rejected bandit. Importantly, those previous models measure a single 'PMFCA' effect across both outcome types and hence do not allow for a separate quantification of MB guidance to MFCA for the informative and non-informative outcomes. Importantly, our current model-agnostic analyses suggested that drug effects on PMFCA might be selective for one outcome type (the informative destination).

We therefore found it necessary to develop more flexible models which would allow for separate quantification of MFCA for each outcome × bandit type. One approach would have been to introduce an MF learning rate per bandit (nominated/rejected) × outcome (informative/non-informative). However, this is problematic because one would then have to run two RW updates per trial per bandit, and where the second update might dilute the effects of the first update. Instead, our current models allow for a more elegant and straightforward solution via the CA parameters. In this approach, only a single value update is executed per bandit, and the CA parameters measure the extent to which a reward for each destination type is assigned to each vehicle. For consistency, we also formulated MB learning using this parameterization.

We discuss in detail the similarities and differences between models based on learning-rate and credit assignment parameterization in a recent paper (*Moran et al., 2021b*) to which we now refer the readers. Interestingly, in that paper we also found that credit-assignment parameters benefit from better estimation properties. We next elaborate more on the similarities and differences between a 'ca parametrization' and the more standard Rescorla-Wagner (RW) 'reward-based parameterization'. At first glance, our value updates may seem very different from standard RW updates. However, it is important to note that these update rules are actually equivalent up to a re-parameterization. To see this, consider a Q-value update of the form we use in our models (henceforth the 'ca parametrization'):

$$Q \leftarrow (1-f) * Q \pm ca \tag{A1}$$

This update can be equivalently written as (henceforth the 'r parameterization'):

$$Q \leftarrow Q + f * \left( \frac{ca}{f} - Q \right) = Q + f * (r - Q) \tag{A2}$$

where r = ± ca/f. *Equation A2* is of the Rescorla-Wagner form, in which f serves as a leaning rate.

Thus, instead of using the 'ca parametrization' we could equivalently have used the 'r parametrization' and estimated free 'r' (rather than 'ca') parameters to quantify the sensitivity of each system (MB and MF) to rewards provided to the various outcome types (i.e., MB sensitivity to rewards on all trials, MF sensitivity to rewards on standard trials, and MF sensitivity to informative and non-informative (destination) rewards for the ghost-nominated and ghost-rejected vehicles).

It is important to note that neither of these formulations renders it possible to identify these parameters (either ca of r) separately from 'decision noise', that is, soft-max temperature parameters. For example, had our model included an MB soft-max temperature parameter, then doubling all the MB ca (or r) parameters and the choice temperature would have maintained model-predictions invariant. Thus, a scaling of the model is mandatory. Because our focus here is on a composition of credit assignment effects, we used the soft-max temperature as a scaling parameter and fixed it to 1. Admittedly this is arbitrary (as any scaling would be) but it is a useful heuristic that allows us to identify credit-assignment parameters for the various outcomes as well as allows us to compare them.

In passing, we think it is helpful and important to point out that a similar scaling problem affects almost all applications of RL models. Consider for example a simple two-arm bandit task wherein each bandit provides a (1£) reward with a Bernoulli probability. Often, a modeller will simply define outcomes as r = 1 (reward) and r = 0 (non-reward), a practice that allows estimating a decision noise parameter (and a learning rate). Critically, this formulation assumes the subjective value of the reward (or non-reward) is equal for all participants. This assumption is likely to be wrong (e.g., 1£ may subjectively worth more to poorer participants) but is commonly used as a form of scaling.

Finally, to test if the current models support our main conclusion from *Moran et al., 2019*, that retrospective MB inference guides MFCA for both the informative and non-informative outcomes, we reanalysed the Moran et al. data using the current novel models and found converging support (*Appendix 1—figure 8*).

**Appendix 1—table 1.** Mixed effects models on model-agnostic choice data from standard trials.

| Name | Estimate | SE | t-Stat | DF | p-Value | Lower CI | Upper CI |
|---|---|---|---|---|---|---|---|
| **MF choice (standard trials)** REPEAT ~ 1 + C*U*DRUG*ORDER + (C + U + DRUG + ORDER | PART) | | | | | | | |
| (Intercept) | 0.34 | 0.06 | 5.55 | 480 | 0.000 | 0.22 | 0.46 |
| C (common) | 0.67 | 0.07 | 9.14 | 480 | 0.000 | 0.53 | 0.82 |
| U (unique) | 1.54 | 0.09 | 17.40 | 480 | 0.000 | 1.36 | 1.71 |
| DRUG | 0.03 | 0.07 | 0.46 | 480 | 0.643 | –0.11 | 0.18 |
| ORDER | 0.07 | 0.07 | 0.91 | 480 | 0.365 | –0.08 | 0.21 |
| C*U | 0.19 | 0.11 | 1.72 | 480 | 0.085 | –0.03 | 0.40 |
| C*DRUG | 0.07 | 0.11 | 0.67 | 480 | 0.500 | –0.14 | 0.29 |
| U*DRUG | 0.06 | 0.11 | 0.56 | 480 | 0.577 | –0.15 | 0.27 |
| C*ORDER | 0.12 | 0.11 | 1.09 | 480 | 0.277 | –0.10 | 0.33 |
| U*ORDER | –0.11 | 0.11 | –0.99 | 480 | 0.321 | –0.32 | 0.11 |
| DRUG*ORDER | –0.25 | 0.25 | –1.02 | 480 | 0.310 | –0.73 | 0.23 |
| C*U*DRUG | 0.14 | 0.22 | 0.64 | 480 | 0.524 | –0.29 | 0.57 |
| C*U*ORDER | 0.13 | 0.22 | 0.59 | 480 | 0.554 | –0.30 | 0.56 |
| C*DRUG*ORDER | –0.02 | 0.29 | –0.06 | 480 | 0.952 | –0.60 | 0.56 |
| U*DRUG*ORDER | –0.18 | 0.35 | –0.51 | 480 | 0.610 | –0.87 | 0.51 |
| C*U*DRUG*ORDER | –0.22 | 0.44 | –0.50 | 480 | 0.618 | –1.08 | 0.64 |
| **MB choice (standard trials)** GENERALIZE ~ C*P*DRUG*ORDER + (C + P + DRUG + ORDER | PART) | | | | | | | |
| (Intercept) | 0.30 | 0.04 | 6.96 | 7177 | 0.000 | 0.22 | 0.38 |
| C (common) | 0.40 | 0.06 | 6.22 | 7177 | 0.000 | 0.27 | 0.52 |
| P (common reward probability) | 1.33 | 0.21 | 6.39 | 7177 | 0.000 | 0.92 | 1.74 |
| DRUG | –0.13 | 0.08 | –1.65 | 7177 | 0.099 | –0.29 | 0.03 |
| ORDER | –0.13 | 0.08 | –1.57 | 7177 | 0.116 | –0.29 | 0.03 |
| C*P | –0.23 | 0.23 | –1.01 | 7177 | 0.311 | –0.67 | 0.21 |
| C*DRUG | 0.05 | 0.12 | 0.39 | 7177 | 0.695 | –0.19 | 0.28 |
| P*DRUG | –0.34 | 0.23 | –1.48 | 7177 | 0.140 | –0.79 | 0.11 |
| C*ORDER | –0.06 | 0.12 | –0.52 | 7177 | 0.606 | –0.30 | 0.17 |
| P*ORDER | 0.16 | 0.23 | 0.70 | 7177 | 0.482 | –0.29 | 0.61 |
| DRUG*ORDER | –0.24 | 0.17 | –1.41 | 7177 | 0.158 | –0.58 | 0.09 |
| C*P*DRUG | –0.08 | 0.45 | –0.18 | 7177 | 0.856 | –0.97 | 0.80 |
| C*P*ORDER | 0.57 | 0.45 | 1.26 | 7177 | 0.207 | –0.31 | 1.45 |
| C*DRUG*ORDER | –0.38 | 0.25 | –1.48 | 7177 | 0.140 | –0.87 | 0.12 |
| P*DRUG*ORDER | 0.46 | 0.83 | 0.55 | 7177 | 0.583 | –1.18 | 2.09 |
| C*P*DRUG*ORDER | 1.40 | 0.91 | 1.54 | 7177 | 0.123 | –0.38 | 3.17 |

**Appendix 1—table 2.** Mixed effects models on model-agnostic choice data from uncertainty trials.

| Name | Estimate | SE | t-Stat | DF | p-Value | Lower CI | Upper CI |
|---|---|---|---|---|---|---|---|
| Preferential MFCA for the informative destination (ghost-nominated, 'repeat trials' > ghost-rejected, 'switch trials')<br>MFCA ~ NOM*DRUG*ORDER + (NOM + DRUG + ORDER \| PART) | | | | | | | |
| (Intercept) | 0.10 | 0.01 | 8.27 | 239 | 0.000 | 0.07 | 0.12 |
| NOM (nomination) | 0.03 | 0.02 | 1.57 | 239 | 0.117 | –0.01 | 0.08 |
| DRUG | 0.01 | 0.02 | 0.40 | 239 | 0.687 | –0.04 | 0.06 |
| ORDER | 0.00 | 0.02 | 0.13 | 239 | 0.895 | –0.04 | 0.05 |
| NOM*DRUG | 0.11 | 0.04 | 2.73 | 239 | 0.007 | 0.03 | 0.19 |
| NOM*ORDER | 0.02 | 0.04 | 0.51 | 239 | 0.613 | –0.06 | 0.10 |
| DRUG*ORDER | –0.03 | 0.05 | –0.73 | 239 | 0.467 | –0.12 | 0.06 |
| NOM*DRUG*ORDER | –0.01 | 0.09 | –0.04 | 239 | 0.966 | –0.18 | 0.17 |
| MFCA for non-informative destination (ghost-nominated > ghost-rejected, 'clash trials')<br>REPEAT ~ N*I*DRUG*ORDER + (N*I*DRUG + ORDER \| PART) | | | | | | | |
| (Intercept) | 0.05 | 0.04 | 1.27 | 479 | 0.203 | –0.03 | 0.12 |
| N (non-informative) | 0.13 | 0.07 | 1.96 | 479 | 0.051 | 0.00 | 0.26 |
| I (informative) | 1.01 | 0.10 | 9.95 | 479 | 0.000 | 0.81 | 1.21 |
| DRUG | 0.16 | 0.07 | 2.31 | 479 | 0.021 | 0.02 | 0.29 |
| ORDER | 0.03 | 0.07 | 0.41 | 479 | 0.684 | –0.10 | 0.16 |
| N*U | 0.08 | 0.14 | 0.57 | 479 | 0.568 | –0.19 | 0.35 |
| N*DRUG | 0.05 | 0.13 | 0.39 | 479 | 0.696 | –0.21 | 0.31 |
| I*DRUG | 0.03 | 0.15 | 0.24 | 479 | 0.810 | –0.25 | 0.32 |
| N*ORDER | –0.05 | 0.13 | –0.34 | 479 | 0.733 | –0.30 | 0.21 |
| I*ORDER | 0.06 | 0.14 | 0.43 | 479 | 0.664 | –0.22 | 0.35 |
| DRUG*ORDER | –0.20 | 0.15 | –1.37 | 479 | 0.171 | –0.49 | 0.09 |
| N*I*DRUG | 0.07 | 0.29 | 0.26 | 479 | 0.798 | –0.49 | 0.64 |
| N*I*ORDER | 0.25 | 0.29 | 0.86 | 479 | 0.388 | –0.32 | 0.81 |
| N*DRUG*ORDER | –0.47 | 0.26 | –1.80 | 479 | 0.072 | –0.99 | 0.04 |
| I*DRUG*ORDER | –0.12 | 0.41 | –0.31 | 479 | 0.759 | –0.92 | 0.67 |
| N*I*DRUG*ORDER | 0.86 | 0.55 | 1.56 | 479 | 0.119 | –0.22 | 1.94 |

**Appendix 1—table 3.** Mixed effects models on parameters of the computational model.

| Name | Estimate | SE | t-Stat | DF | p-Value | Lower CI | Upper CI |
|---|---|---|---|---|---|---|---|
| MFCA for ghost-nominated vs. ghost-rejected and informative vs. non-informative<br>MFCA ~ NOM*INFO*DRUG*ORDER + (NOM*INFO*DRUG + ORDER \| PART) | | | | | | | |
| (Intercept) | 0.18 | 0.02 | 7.60 | 480 | 0.000 | 0.14 | 0.23 |
| NOM (nomination) | 0.10 | 0.03 | 3.72 | 480 | 0.000 | 0.05 | 0.15 |
| INFO (informativeness) | 0.08 | 0.04 | 2.19 | 480 | 0.029 | 0.01 | 0.15 |
| DRUG | 0.05 | 0.05 | 0.94 | 480 | 0.347 | –0.05 | 0.16 |
| ORDER | 0.04 | 0.05 | 0.73 | 480 | 0.463 | –0.06 | 0.15 |
| NOM*INFO | –0.03 | 0.05 | –0.57 | 480 | 0.567 | –0.12 | 0.06 |
| NOM*DRUG | 0.10 | 0.04 | 2.43 | 480 | 0.015 | 0.02 | 0.18 |

*Appendix 1—table 3 Continued on next page*

*Appendix 1—table 3 Continued*

| Name | Estimate | SE | t-Stat | DF | p-Value | Lower CI | Upper CI |
|------|----------|-----|--------|-----|---------|----------|----------|
| INFO*DRUG | –0.08 | 0.07 | –1.16 | 480 | 0.247 | –0.22 | 0.06 |
| NOM*ORDER | 0.02 | 0.04 | 0.37 | 480 | 0.715 | –0.07 | 0.10 |
| INFO*ORDER | 0.10 | 0.07 | 1.42 | 480 | 0.157 | –0.04 | 0.23 |
| DRUG*ORDER | –0.09 | 0.10 | –0.98 | 480 | 0.328 | –0.28 | 0.10 |
| NOM*INFO*DRUG | 0.02 | 0.07 | 0.33 | 480 | 0.738 | –0.12 | 0.17 |
| NOM*INFO*ORDER | –0.01 | 0.07 | –0.08 | 480 | 0.934 | –0.15 | 0.14 |
| NOM*DRUG*ORDER | –0.06 | 0.11 | –0.60 | 480 | 0.551 | –0.27 | 0.15 |
| INFO*DRUG*ORDER | 0.16 | 0.14 | 1.10 | 480 | 0.272 | –0.12 | 0.44 |
| NOM*INFO*DRUG*ORDER | 0.10 | 0.19 | 0.55 | 480 | 0.585 | –0.26 | 0.47 |

Preferential MFCA for informative vs. non-informative
PMFCA ~ INFO*DRUG*ORDER + (INFO + DRUG + ORDER | PART)

| Name | Estimate | SE | t-Stat | DF | p-Value | Lower CI | Upper CI |
|------|----------|-----|--------|-----|---------|----------|----------|
| (Intercept) | 0.10 | 0.03 | 3.71 | 240 | 0.000 | 0.05 | 0.15 |
| INFO (informativeness) | –0.03 | 0.05 | –0.57 | 240 | 0.568 | –0.12 | 0.07 |
| DRUG | 0.10 | 0.04 | 2.39 | 240 | 0.017 | 0.02 | 0.18 |
| ORDER | 0.02 | 0.04 | 0.36 | 240 | 0.720 | –0.07 | 0.10 |
| INFO*DRUG | 0.02 | 0.07 | 0.34 | 240 | 0.734 | –0.12 | 0.17 |
| INFO*ORDER | –0.01 | 0.07 | –0.08 | 240 | 0.933 | –0.15 | 0.14 |
| DRUG*ORDER | –0.06 | 0.11 | –0.60 | 240 | 0.552 | –0.27 | 0.15 |
| INFO*DRUG*ORDER | 0.10 | 0.19 | 0.55 | 240 | 0.585 | –0.27 | 0.47 |

**Appendix 1—table 4.** Distribution of parameters from the full computational model.

| Cond. | % | MFCA standard | MFCA info-nom | MFCA info-rej | MFCA non-info-nom | MFCA non-info-rej | MBCA | Perseveration-standard | perseveration-nominated | forget_MF | forget_MB | forget_Pers |
|-------|-----|----------|----------|----------|----------|----------|------|----------|----------|----------|----------|----------|
| | 25 | 0.053 | –0.056 | –0.026 | –0.070 | –0.074 | 0.059 | –0.197 | –0.093 | 0.002 | 0.038 | 0.010 |
| | 50 | 0.147 | 0.168 | 0.149 | 0.048 | 0.030 | 0.273 | 0.042 | 0.071 | 0.058 | 0.148 | 0.123 |
| Placebo | 75 | 0.364 | 0.479 | 0.391 | 0.333 | 0.204 | 0.454 | 0.383 | 0.353 | 0.519 | 0.521 | 0.428 |
| | 25 | 0.060 | –0.025 | –0.073 | –0.011 | –0.098 | 0.026 | –0.086 | –0.047 | 0.019 | 0.022 | 0.008 |
| | 50 | 0.272 | 0.165 | 0.130 | 0.178 | 0.070 | 0.278 | 0.098 | 0.084 | 0.190 | 0.127 | 0.089 |
| Levodopa | 75 | 0.574 | 0.517 | 0.383 | 0.390 | 0.291 | 0.367 | 0.346 | 0.374 | 0.598 | 0.508 | 0.492 |

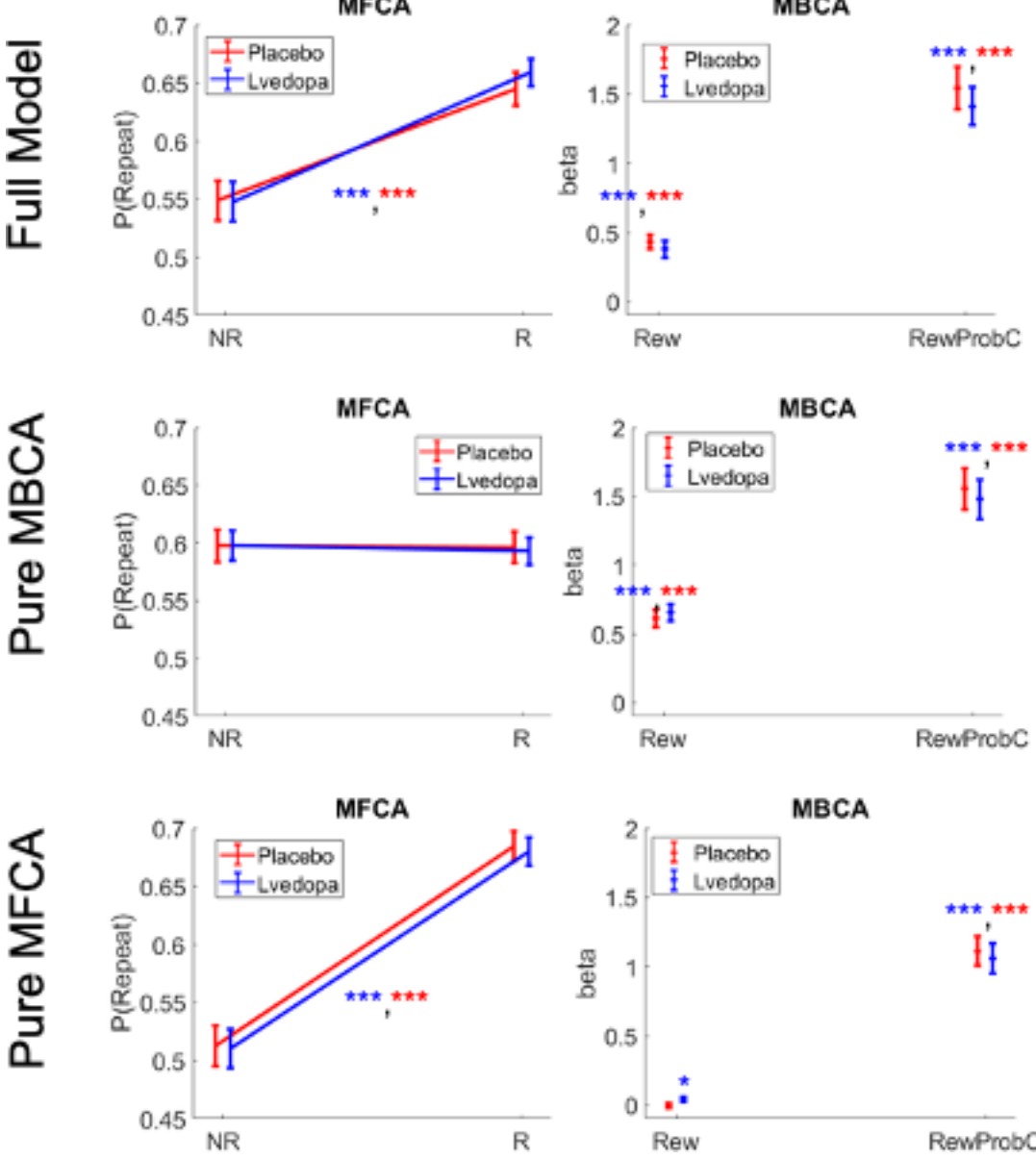

**Appendix 1—figure 1.** Simulations for standard trials based on the full model and sub-models. NR = no reward, R = reward. Rew = reward at the common destination, RewProBC = reward probability at the common destination.

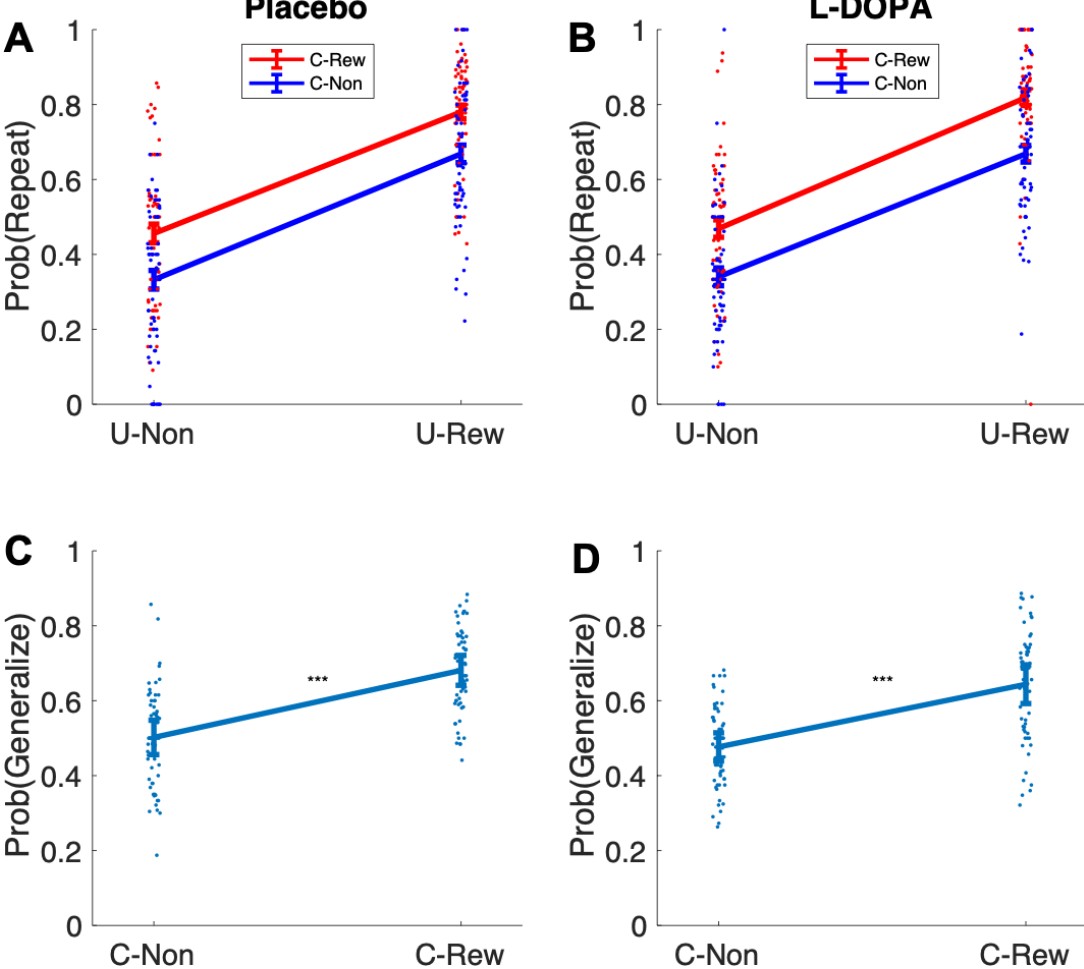

**Appendix 1—figure 2.** Empirical probabilities of model-agnostic model-free (MF) (A and B) and model-based (MB) (C and D) choice contribution under placebo and levodopa (L-Dopa). U-Non = no reward at unique destination, U-Rew = reward at unique destination, C-Non = no reward at common destination, C-Rew = reward at common destination.

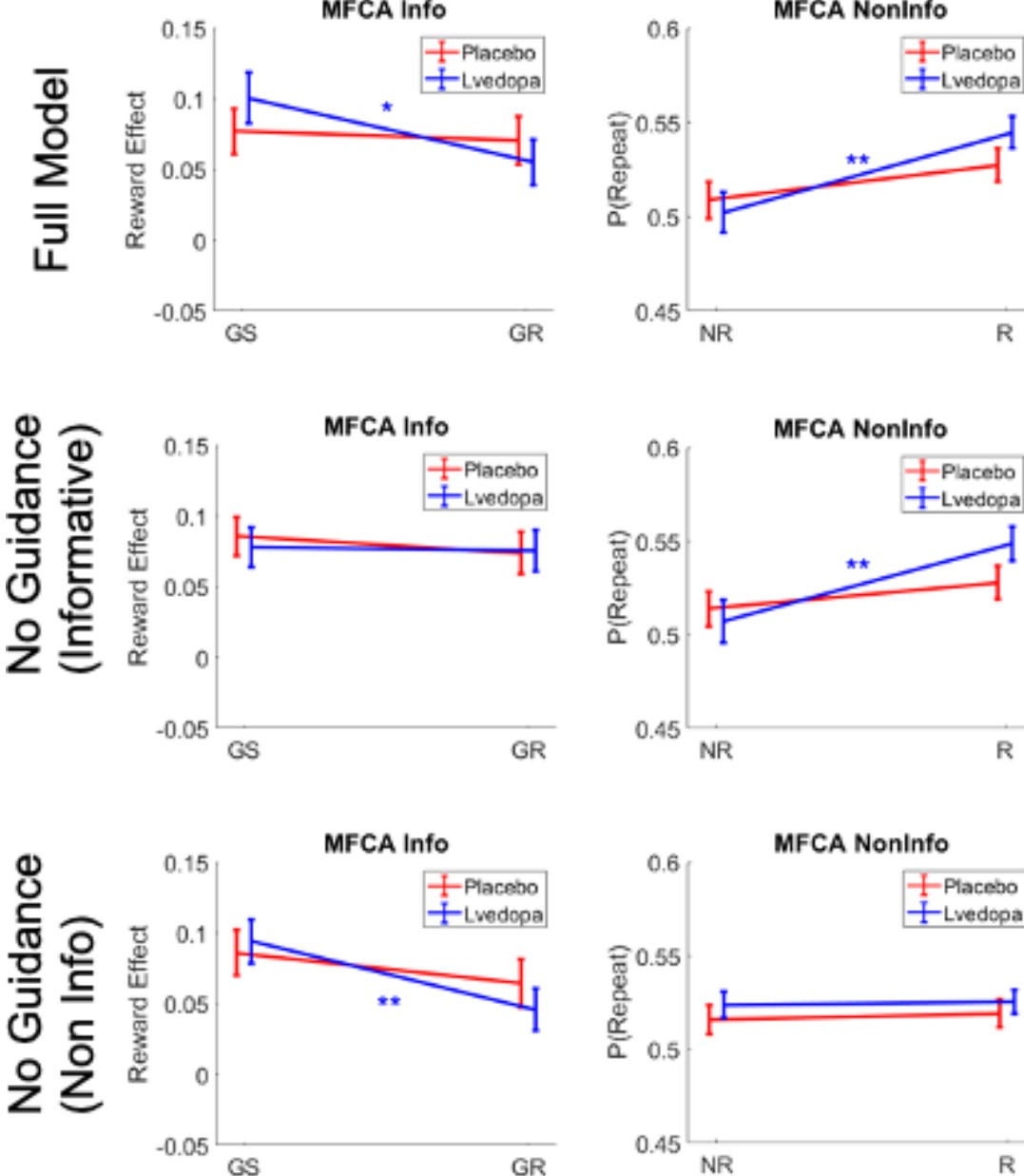

**Appendix 1—figure 3.** Simulations for uncertainty trials based on the full model and sub-models. GS = ghost-selected, GR = ghost-rejected.

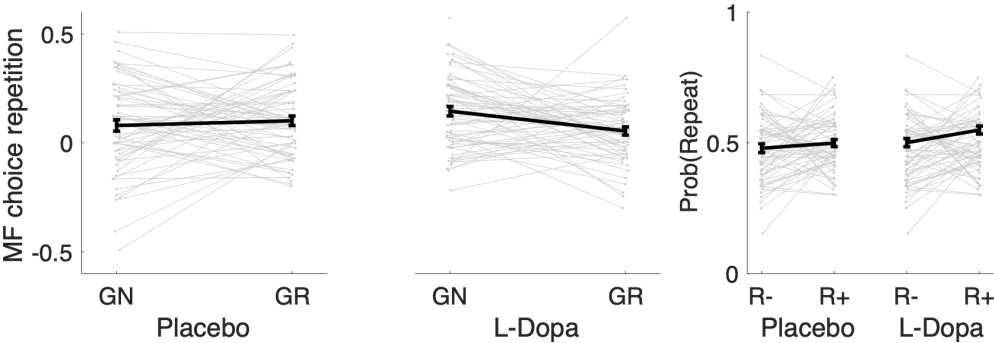

**Appendix 1—figure 4.** Same data as plotted in *Figure 4* in the main manuscript but individual variability reflects differences in task conditions. GN = ghost-nominated, GR = ghost-rejected, R− = no reward, R+ = reward.

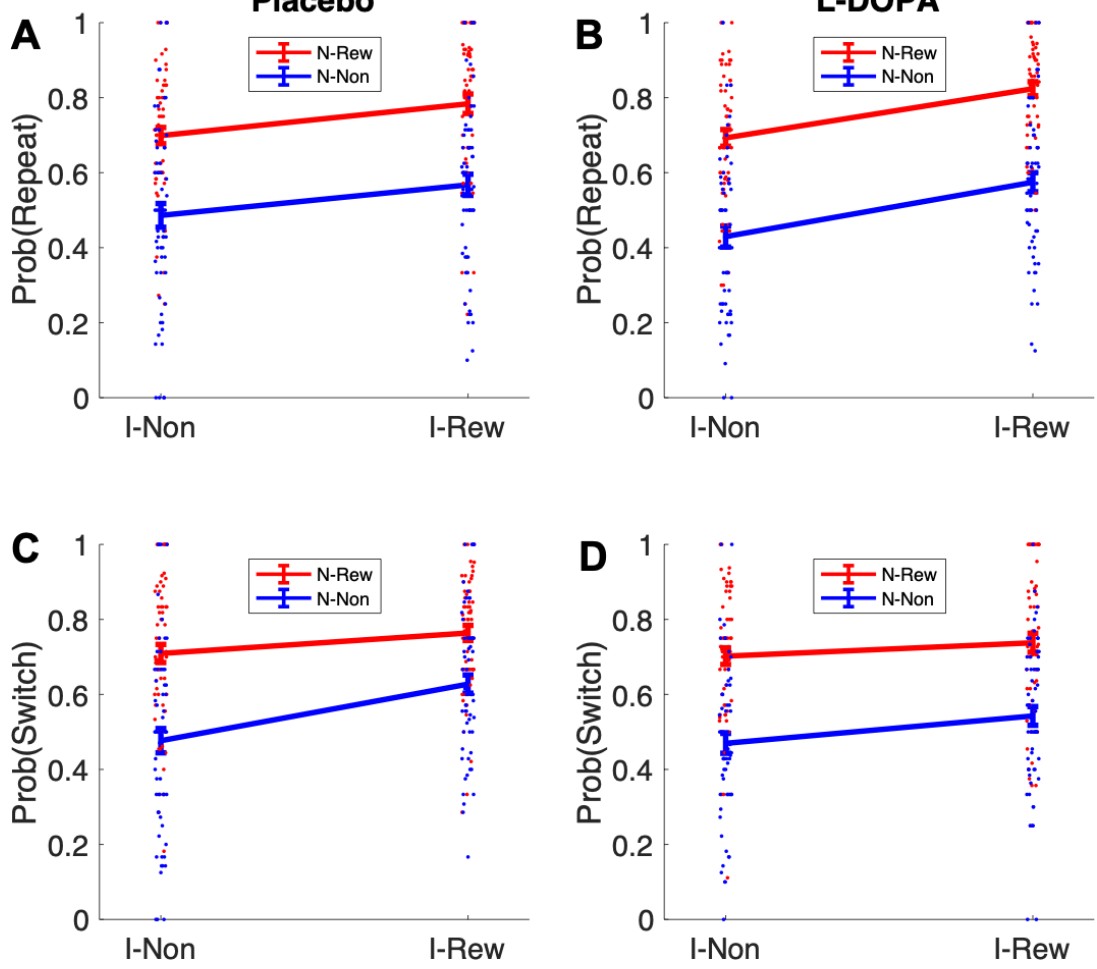

**Appendix 1—figure 5.** Retrospective model-based (MB) inference using the informative destination based on repeat and switch signatures after uncertainty trials. I-Non = no reward at informative destination, I-Rew = reward at informative destination, N-Non = no reward at non-informative destination, N-Rew = reward at non-informative destination.

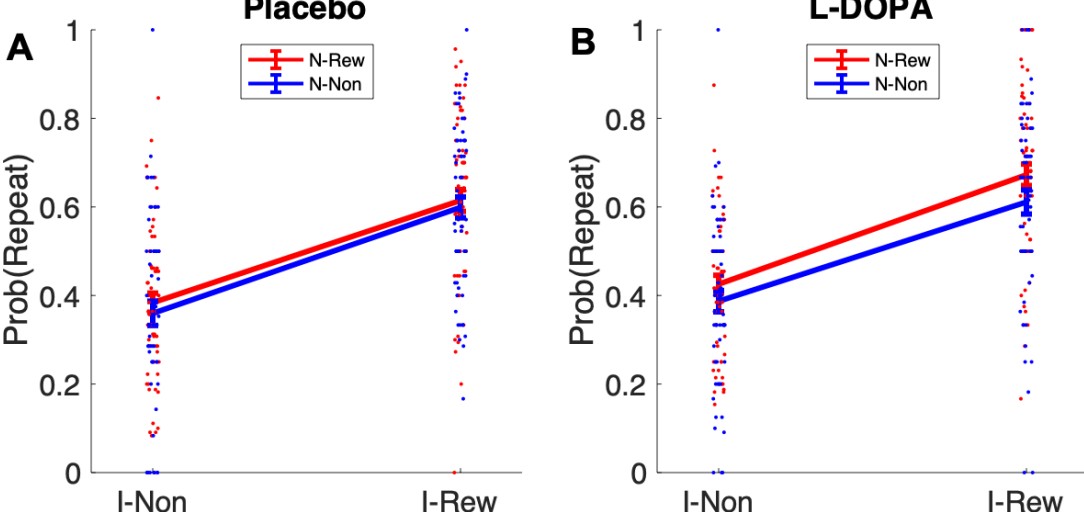

**Appendix 1—figure 6.** Retrospective model-based (MB) inference using the non-informative destination based on choice repetition in 'clash' trials n + 1 following an uncertainty trial n. I-Non = no reward at informative destination, I-Rew = reward at informative destination, N-Non = no reward at non-informative destination, N-Rew = reward at non-informative destination.

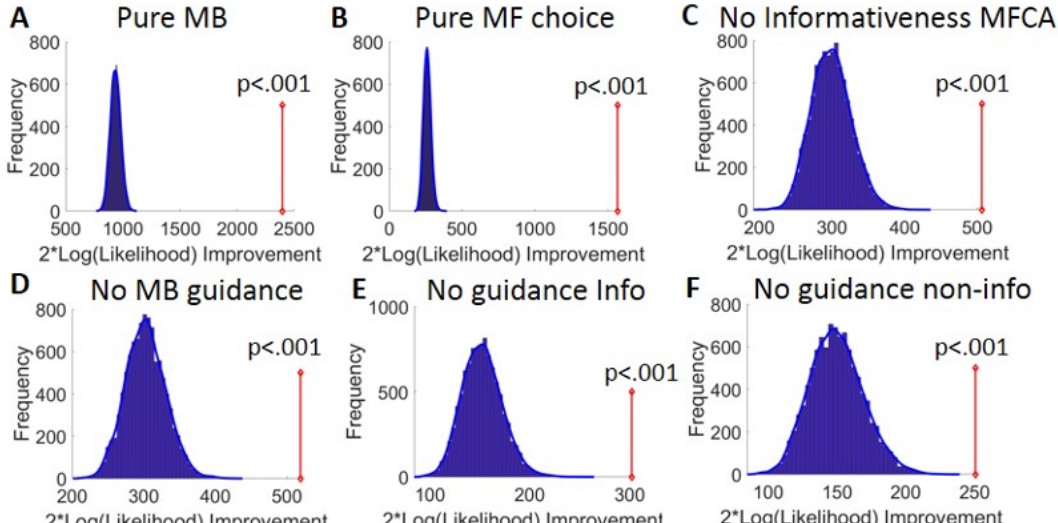

**Appendix 1—figure 7.** Model comparison results. (**A**) Results of the bootstrap-GLRT model comparison for the pure model-based (MB) sub-model. The blue bars show the histogram of the group twice log-likelihood improvement (model vs. sub-model) for synthetic data simulated using the sub-model (10,000 simulations). The blue line displays the smoothed null distribution (using Matlab's 'ksdensity'). The red line shows the empirical group twice log-likelihood improvement. p-Value reflects the proportion of 10,000 simulations that yielded an improvement in likelihood that was at least as large as the empirical improvement. (**B–E**) Same as (**A**), but for the pure model-free (MF) choice, the no informativeness effects on MF credit assignment (MFCA), the no MB guidance for MFCA, the no MB guidance for the informative destination, and the no MB guidance for the non-informative destination sub-models.

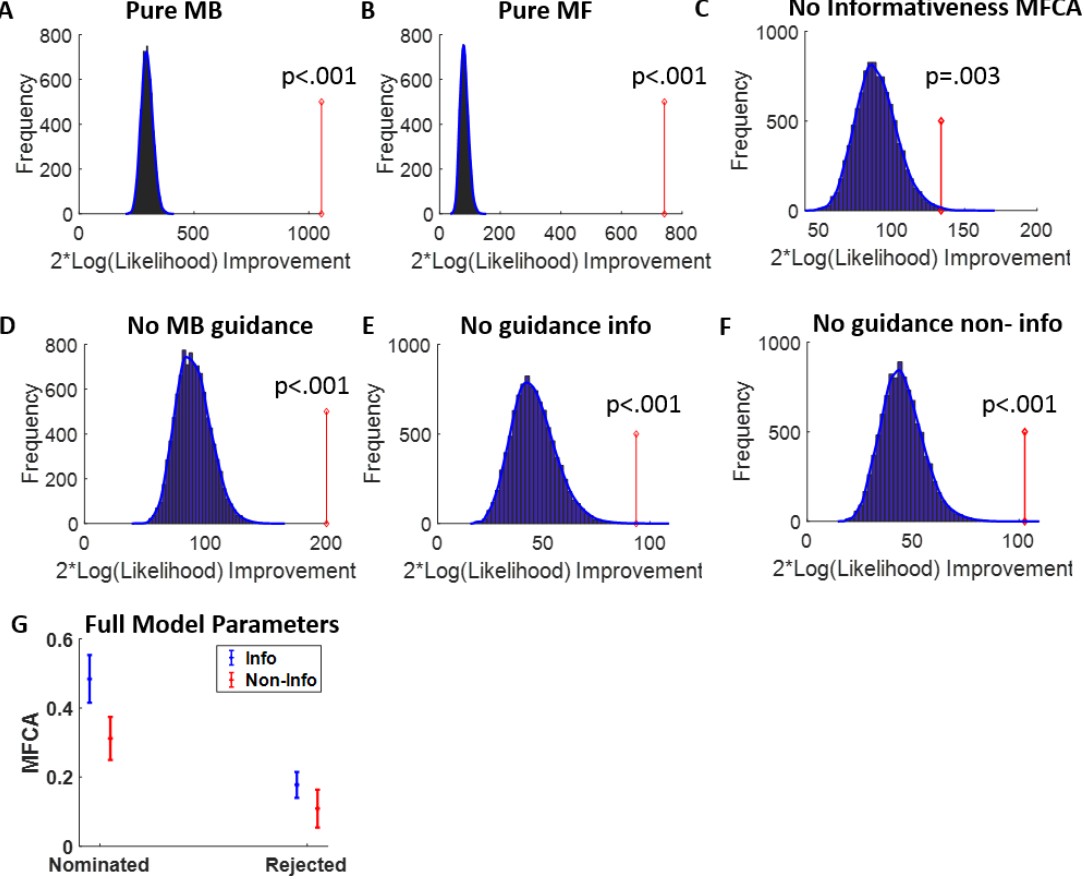

**Appendix 1—figure 8.** Reanalysis of *Moran et al., 2019*, based on the current models. (**A–F**) As *Appendix 1—figure 7* but for the data of *Moran et al., 2019*. Each of the sub-models was rejected at the group level in favour of the full model. (**G**) Full-model uncertainty trials model-based credit assignment (MFCA) parameters as a function of outcome informativeness (blue/red) and nomination. Using a mixed effects model, we found a significant interaction effect between informativeness and nomination (b = 0.10, t = 2.05, p = 0.042) implying the nomination effect on MFCA was stronger for the informative than the non-informative outcome. Simple effect analysis showed significant positive nomination effects for both the informative outcome (blue; b = 0.2, F(1,156) = 28.16, p = 4e-7) and the non-informative outcome (red; b = 0.31, F(1,156) = 12.36, p = 6e-4). Thus, this analysis supports the conclusions from *Moran et al., 2019*, that retrospective model-based (MB) inference guides MFCA on uncertainty trials for both outcomes. Note that *Moran et al., 2019*, did not separate MFCA for the informative and non-informative outcomes.

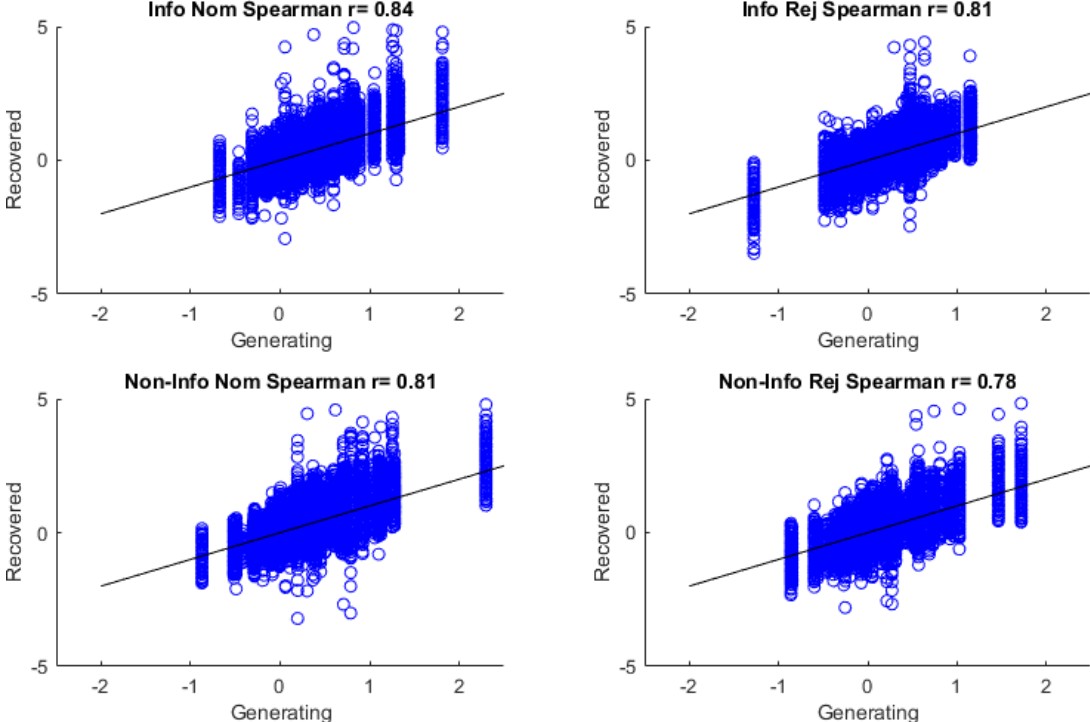

**Appendix 1—figure 9.** Parameter recoverability. For each of the 2*62 full-model parameter combinations, 1000 synthetic (simulated) datasets were created by simulating the full model on experimental sessions as in the true experiment. Then the full model was fit to each of these generated datasets. For each credit assignment (CA) parameter we plot the recovered against the generating parameters, report the Spearman correlation and impose black diagonals where 'recovered = generating'. (**A**) Model-free CA (MFCA) on standard trials, (**B–E**) MFCA on uncertainty trials; (**B**) informative outcome, ghost-nominated, (**C**) informative outcome, ghost-rejected, (**D**) non-informative outcome, ghost-nominated, (**E**) non-informative outcome, ghost-rejected, (**F**) model-based CA (MBCA).

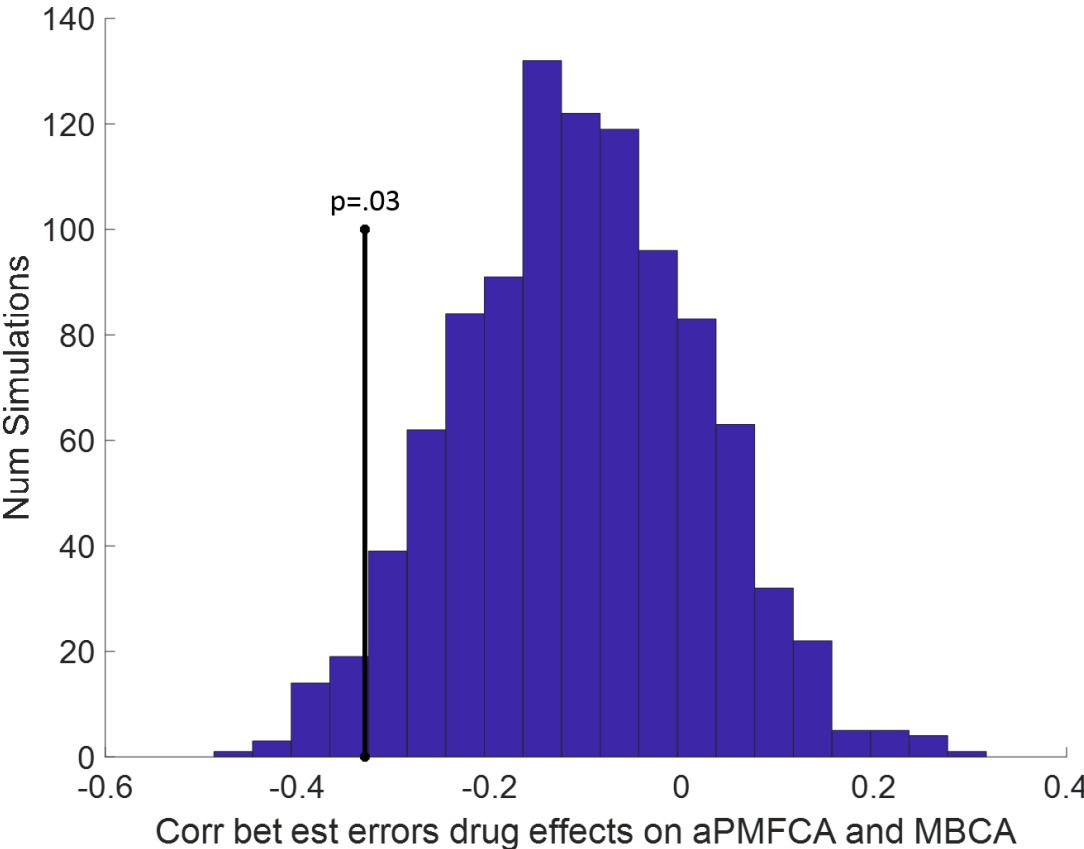

**Appendix 1—figure 10.** Trade-off between parametric drug effects on averaged preferential model-free credit assignment (aPMFCA) and model-based credit assignment (MBCA). Based on our parameter recovery simulations (see *Appendix 1—figure 9*), we also calculated for each participant and each simulation estimation errors (est errors) for drug effects on aPMFCA and MBCA (as differences between fitted and generating drug effects). Next for each simulation index (i = 1,2,…,1000) we calculated the group-level Spearman correlation between these two estimation errors. The histogram of these correlations is plotted. There is weak negative trade-off between estimation errors of drug effects on aPMFCA and MBCA. Importantly, the negative empirical correlation (vertical black line) was still significant even after controlling for this trade-off (p = 0.03; calculated as the proportion of simulations with correlation ≤ empirical correlation).

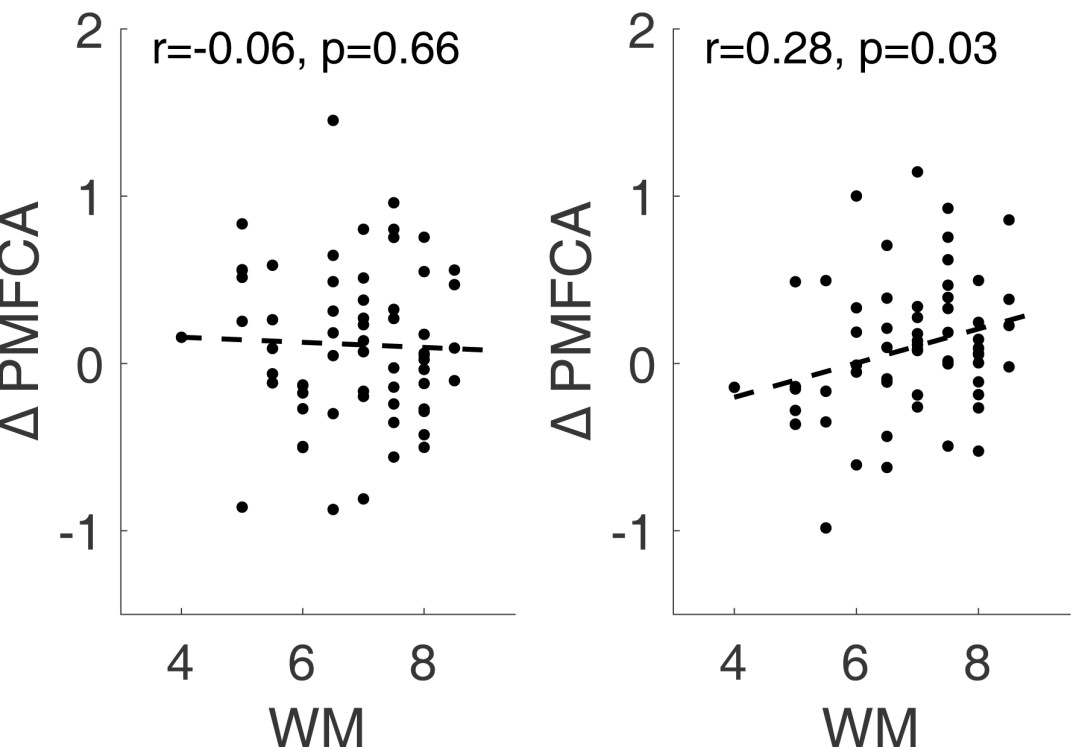

**Appendix 1—figure 11.** Scatter plots of the drug effect (levodopa minus placebo) on preferential model-free credit assignment (ΔPMFCA) based on the informative destination reward and for the non-informative destination reward against working memory (WM).

