## [Editor Report]

This behavioral pharmacology study provides convincing evidence that dopamine governs interactions between model-based and model-free learning systems. The issue addressed is timely and clinically relevant, and will be of interest to a broad audience interested in dopamine, learning, choice, and planning.

---

## [Decision Letter]

**Decision letter after peer review:**

Thank you for submitting your article "Dopamine enhances model-free credit assignment through boosting of retrospective model-based inference" for consideration by *eLife*. Your article has been reviewed by 3 peer reviewers, one of whom is a member of our Board of Reviewing Editors, and the evaluation has been overseen by Christian Büchel as the Senior Editor. The following individual involved in review of your submission has agreed to reveal their identity: Roshan Cools (Reviewer #3).

Essential revisions:

All reviewers agreed that this is an important study with interesting results. Reviewers specifically commended the clever task design and the rigor of the experimental approach. There were, however, also several concerns that we would ask you to address in a revision.

1. There was initial disagreement among reviewers as to whether the results provide convincing evidence for effects of levodopa on retrospective model-based (MB) inference on model-free (MF) credit assignment (CA). An extensive discussion revealed that this disagreement depended on whether an interpretation of what trials were included in the "clash condition". Namely, whether the clash condition includes "any" standard trials in which both ghost- nominated (GN) and ghost-rejected (GR) vehicles are presented, regardless of whether the reward in the preceding uncertainty trial was given at the informative or the non-informative destination. In this case, the clash condition would be the relevant condition to isolate guidance of MFCA by retrospective MB inference. On the other hand, if the clash condition consists only of trials where reward is presented at the non-informative destination, then the clash condition is not the primary test of this, but then it would be unclear why this condition is so prominently discussed in the main text and in Figure 3.

2. Related to this, it was unclear how many trials (on average per subject) went into each of the different experimental conditions/analyses in Figure 3 A1, A2, C, and in Figure 2 A and C. This is important and should be mentioned in the main text.

3. The disagreement on how to interpret what is shown in the figures and what analyses were conducted underscores a concern that was mentioned by several reviewers. Namely, the manuscript is difficult to understand and we encourage you to re-work the presentation of the task, the logic of the analyses, and what effects were expected under the different conditions (and why). Related to this, one reviewer noted that there is an error in Figure 3C, such that for standard trial t+1, the yellow vehicle should be paired with the forest and the desert destination, not the city and beach.

4. As you can see in the individual comments, several reviewers raised concerns about the individual differences analyses. They appeared surprising, post-hoc, and not always easy to interpret. Reviewers noted that these concerns can be addressed by reducing emphasis on, or even removing, these individual differences results from the main text.

5. Reviewers noted the lack of effects of levodopa on MF and MB learning. While this is something that has been seen previously, it also might reflect lack of sensitivity of the manipulation with the current task. It would be important to elaborate on the implications of the lack of basic MF and MB modulation.

6. Relatedly, the critical MB guidance of MFCA effect was not observed under placebo (Figure 3 B). The computational model-based analyses do speak to this issue, but are the results from the model-informed and model-agnostic analyses otherwise consistent? Model-agnostic analyses reveal a greater effect of levodopa on informative destination for the GN than the GR trials and no effect for non-informative destination. Conversely model-informed analyses reveal a nomination effect of levodopa across informative and non-informative trials. This was not addressed. Also, how does this model space, the winning model and the modeling exercise differ (or not) from that in the previous paper by Moran et al. without levodopa?

7. What putative neural mechanism led the authors to predict this selective modulation of the interaction? The introduction states that "Given DA's contribution to both MF and MB systems, we set out to examine whether this aspect of MB-MF cooperation is subject to DA influence." This is vague. For the hypothesis to be plausible, it would need to be grounded in some idea about how the effect would be implemented. Where exactly does dopamine act to elicit an effect on retroactive MB inference, but not MB learning? If the mechanism is a modulation of working memory and/or replay itself, then shouldn't that lead to boosting of both MB learning as well as MB influences on MF learning? Addressing this involves specification of the mechanistic basis of the hypothesis in the introduction, but the question also pertains to the Discussion section. Hippocampal replay is invoked, but can the authors clarify why a prefrontal working memory (retrieval) mechanism invoked in the preceding paragraph would not suffice. In any case, it seems that an effect of dopamine on replay would also alter MB planning.

8. The critical drug effect seems marginally significant and the key plots in Figure 3B and Figure 4 B, C and D do not show relevant variability in the drug effect. It would be important to connect (or otherwise plot) individual data points between the levodopa and placebo conditions, allowing readers to appreciate the relevant individual variability in the drug effects.

9. Please provide a rationale for changing the stimuli and details of the task relative to what was used in Moran et al.

10. The amount of training depended on performance on a quiz. How much training did people receive and was this matched between drug sessions?

11. Please include a note on the implications of this study to psychiatric disorders that are related to dopamine.

12. In addition, individual reviewers made a number of suggestions to improve the manuscript. Please consider these additional points when preparing the revised manuscript.

*Reviewer #1 (Recommendations for the authors):*

1. It would be important to expand the discussion of two somewhat surprising findings. First, please expand on why individual differences in the effect of dopamine on MB inference would negatively correlate with the effects of retrospective MB inference on MFCA. Second, please elaborate on why you think the current results do not replicate previously reported effects of dopamine on MB learning.

2. The logic of the analysis is complex. It took me a while to figure out what the critical effects are Figure 3. You may want to consider revising the presentation of their analyses, if possible. Could you expand the schematics in the figures? Perhaps you could add a flow chart of how the different effects are isolated. I realize it may be difficult to come up with this, but I believe it would make your findings much more accessible.

*Reviewer #2 (Recommendations for the authors):*

– I believe that the most important effect in this study is the one presented in Figure 3C and 3D, which authors have called the "Clash condition". In this one, the same chosen pair on the preceding uncertainty trial is presented in a standard trial and subjects are asked to choose between the two choices. This is, I believe, is the ultimate test trials in the study; and there is no significant effect of drug in those trials. Looking at Table S2, it seems to me that authors have done a very good job in increasing the within-subject power for that trial type (mixed-effect df for the clash trials is 4861; the df for the repeat/switch trials is 239 according to Table S2). Related to this point, authors have found no significant effect of DA on credit assignment in their computational modeling analysis.

If this is correct (and I am happy to be convinced otherwise), then I have two suggestions. First, authors should do a Bayesian test to test if the null hypothesis (no drug effect) is significantly preferred to the alternative hypothesis. If this is the case, I think the study is very informative (but actually in the opposite direction of the current claim). Second, if the Bayesian test does not prefer the null hypothesis, then I suggest to revise the title/abstract and conclusion part of the study; and simply state that there was no conclusive effect of DA on credit assignment. In this case, I think the other part of results (i.e Figure 3B) as well as the span-dependent effect on parameters from the model-based analyses (Figure 5) are of interest and significance.

– The part of discussion about the hippocampal replay comes across as highly speculative and not grounded in the current findings. I suggest to instead focus on limitations of the current study.

– Labels are not matched with panels in Figure 4.

– I suggest to move Figure 2 to the supplementary, or at least present it after Figure 3. This figure is not about the credit assignment, which is the main focus of this study.

– I don't think that authors statement of Cools' theory in terms of DA synthesis capacity is correct (and it is a bit unfair too) (page 17). I believe the main prediction, based on that theory, or at least its recent form, is that DA effect is baseline-dependent and therefore it is actually quite consistent with what authors found. Based on this theory, WM-span test is a good "proxy" of baseline dopamine synthesis capacity. I suggest to revise that part of the discussion.

– I believe that a note on how this study might be related to psychiatric disorders that are related to dopamine would be of interest for many readers.

*Reviewer #3 (Recommendations for the authors):*

It would be good make explicit exactly how many trials there were for the various conditions of interest: e.g. MF choice repetition trials, MB choice generalization trials and critical retrospective MB inference trials.

Figure 3C: is there a mistake in the destinations paired with the yellow car on the standard trial n+1?

Please provide a rationale for changing the stimuli and details concerning shortening the task cf Moran et al.

The amount of training depended on performance on a quiz. How much training did people receive and was this matched between drug sessions?

While I consider myself somewhat of an expert, I found the paper really quite difficult to parse, in part due to the large number of abbreviations, constructs and interactions invoked. So I am somewhat concerned that it will also be hard to appreciate for a somewhat broader audience. One thing that can be done is to be more consistent in the use of relevant labels:

For example, ghost trials vs uncertainty trials

PMBCA vs retrospective MB inference

[Editors' note: further revisions were suggested prior to acceptance, as described below.]

Thank you for resubmitting your work entitled "Dopamine enhances model-free credit assignment through boosting of retrospective model-based inference" for further consideration by *eLife*. Your revised article has been reviewed by 3 peer reviewers, one of whom is a member of our Board of Reviewing Editors, and the evaluation has been overseen by Christian Büchel as the Senior Editor.

The manuscript has been improved but there are some remaining issues that need to be addressed, as outlined below:

All reviewers thought that the revised manuscript adequately addressed most of their initial concerns and improved the clarity and presentation of your work. There was only one remaining concern by Review #2, which we would like you to address before we make a final decision about your manuscript.

*Reviewer #1 (Recommendations for the authors):*

The authors have adequately addressed my initial comments. I have no further questions.

*Reviewer #2 (Recommendations for the authors):*

In this revision, Deserno and colleagues have revised the original manuscript extensively, especially regarding presentation of the task, analyses and findings. And I am thankful to authors for doing that.

I still think that the clash condition is the more intuitive measure for CA in this task. But I am fine with the way that authors present and interpret their findings. One pressing issue, however, is still unresolved for me and I hope that authors can clarify that. In the previous revision, I asked about number of trials went into each condition. The main reason I asked that was because the df for the switch and repeat trials was very small given for a mixed effect within-subject analysis (i.e. 239, page 14). By examining the manuscript more thoroughly, I believe that the main reason is that authors have conducted a different type of analysis for those trials compared to the rest of manuscript. In particular, analysis of repeat and switch trials are based on proportion of choice (page 33.) This is quite different from the other analyses. What is the rationale for not doing a logistic regression here? Does a logistic regression with the same factors result in similar results? Proportion of binary dependent variable is not a good measure statistically. I might miss something here, but I believe clarifying this point makes the main results more convincing.

*Reviewer #3 (Recommendations for the authors):*

I have now carefully read the rebuttal and revision, noted the extensive changes that were made, to the intro, results and Discussion sections, and believe the paper has become more accessible and the analyses more clearly justified as a result. I have no further comments.

---

## [Author Response]

Essential revisions:All reviewers agreed that this is an important study with interesting results. Reviewers specifically commended the clever task design and the rigor of the experimental approach. There were, however, also several concerns that we would ask you to address in a revision.1. There was initial disagreement among reviewers as to whether the results provide convincing evidence for effects of levodopa on retrospective model-based (MB) inference on model-free (MF) credit assignment (CA). An extensive discussion revealed that this disagreement depended on whether an interpretation of what trials were included in the "clash condition". Namely, whether the clash condition includes "any" standard trials in which both ghost- nominated (GN) and ghost-rejected (GR) vehicles are presented, regardless of whether the reward in the preceding uncertainty trial was given at the informative or the non-informative destination. In this case, the clash condition would be the relevant condition to isolate guidance of MFCA by retrospective MB inference. On the other hand, if the clash condition consists only of trials where reward is presented at the non-informative destination, then the clash condition is not the primary test of this, but then it would be unclear why this condition is so prominently discussed in the main text and in Figure 3.

We apologise that our description of this critical, if slightly complicated, condition was not sufficiently clear. In short, as implied, the ‘clash’ trials comprise all cases in which the ghost-nominated and ghost-rejected vehicles from trial n are pitted against each other on trial n+1, irrespective of reward for non-informative or informative destinations on trial n.

In fact, we distinguish MB influence over choice from MB influence over MFCA (and MF choice) in two structurally different ways. Clash trials allow us to make this distinction based on the non-informative destination on trial n. The comparison between ‘repeat’ and ‘switch’ trials n+1 allows us to make the distinction based on the informative destination on trial n. We now include a more thorough description of clash trials (p. 15) as well as revise our description of repeat and switch trials (p. 12/13) to improve clarity.

For clash trials, p. 15:

“A second means to examine MB influences over MFCA is to consider the non-informative destination. […] Note again that this guidance can only be retrospective, since it is only when the highway is presented on trial n that the MB system can infer that the ghost had in fact nominated the green antique car.”

For repeat and switch trials (p.12/13):

“In a “repeat” trial (Figure 4 A1), the vehicle chosen by the ghost (the green antique car) is pitted against another vehicle from the non-chosen uncertainty trial n pair (the blue crane), which shares the informative destination (the highway). […] Note that this guidance of preferential MFCA can only take place once state uncertainty is resolved and therefore by definition can only be retrospective.”

In sum, the different conditions are optimized to capture different aspects in guidance of MFCA by retrospective MB inference, as compared to pure MB evaluations, based upon the informative destination (repeat vs. switch trials) and the non-informative destination (clash trials). In line with point 3, we have thoroughly re-worked the description of the task’s logic and all conditions to improve accessibility of the text.

2. Related to this, it was unclear how many trials (on average per subject) went into each of the different experimental conditions/analyses in Figure 3 A1, A2, C, and in Figure 2 A and C. This is important and should be mentioned in the main text.

Each session (drug as well as placebo) had 360 trials, resulting in 720 trials in total. One third of trials (120 per session) were standard trials (following standard trials) and included 30 presentations of each of four eligible pairs of vehicles in random order. Half of these trials are used for each of the analyses of MF and MB signatures (Figure 2). This resulted in a maximum of 60 trials for the analysis of each of the MF and MB signatures. For each session, every third trial was an uncertainty trial (120 trials), which were always followed by a standard trial, where a third of these trials (40 trials) contributed to each of the “repeat”, “switch” and “clash” conditions (now Figure 4). Accounting for missed choices, this resulted in the following trial averages per condition and per session (placebo/drug): MF-placebo 58.07, MF-drug 59.15; MB-placebo 58.00, MB-drug 57.01; repeat-placebo 39.42, repeat-drug 39.31; switch-placebo 39.34, switch-drug 39.10; clash-placebo 39.34, clash-drug 39.32.

This information is now added to the methods section on p. 30. Of course, the parameters of the computational model are formally fit on all trials.

3. The disagreement on how to interpret what is shown in the figures and what analyses were conducted underscores a concern that was mentioned by several reviewers. Namely, the manuscript is difficult to understand and we encourage you to re-work the presentation of the task, the logic of the analyses, and what effects were expected under the different conditions (and why). Related to this, one reviewer noted that there is an error in Figure 3C, such that for standard trial t+1, the yellow vehicle should be paired with the forest and the desert destination, not the city and beach.

We appreciate this concern. We have substantially amended the following sections to improve accessibility for a less specialized readership: we have reworked the presentation of the task’s logic underlying of MF and MB credit assignment (p. 6-7, subheading “Logic of the dissociation between model-free and model-based credit assignment”) as well as the paragraph introducing the logic of guidance of MFCA by retrospective MB inference (p. 10-12, subheading “Retrospective MB inference guides MFCA”). Furthermore, we commence each of the results sub-sections reporting task and drug effects from a certain condition (p.7/8, subheading “No evidence of dopaminergic modulation for MF choice”; p. 8-10, subheading “No evidence of dopaminergic modulation for MB choice”; p. 12/13, subheading “Dopamine enhances preferential guidance of MFCA at the informative destination”; p. 15/16, subheading “Dopaminergic modulation of preferential MFCA at the non-informative destination”) with a more detailed explanation of the effects expected based on MF and MB evaluations, hopefully linking this better to the logic of the analyses.

We thank the reviewer for detecting the error in the previous Figure 3, which is now corrected as part of the new Figure 4. We apologize if this contributed to the confusion about the rationale of the various conditions.

Following a suggestion by reviewer 1, we now include flow charts that illustrate the conditions and the modelling in Figure 2 (panel A, MFCA vs. MBCA) and a new Figure 3 (pure MBCA versus guidance of MFCA by MB inference).

4. As you can see in the individual comments, several reviewers raised concerns about the individual differences analyses. They appeared surprising, post-hoc, and not always easy to interpret. Reviewers noted that these concerns can be addressed by reducing emphasis on, or even removing, these individual differences results from the main text.

We gratefully appreciate this critique about the analysis of inter-individual differences. In sum, we accept the post-hoc concern and now move the inter-individual differences as they relate to working memory to Appendix 1 (p. 44/45, subheading “Drug effect on guidance of MFCA and working memory”, Appendix 1 – Figure 11). Furthermore, we motivate the exploratory analysis of inter-individual differences in drug effects between guidance of MFCA by retrospective MB inference and MBCA more clearly in the introduction (p. 3/4), based on a previous literature reporting a positive link between DA and MB control (Deserno et al., 2015a; Groman et al., 2019; Sharp et al., 2016; Wunderlich et al., 2012). We have also simplified the analysis of drug-related inter-individual differences between guidance of MFCA by retrospective MB inference and MBCA so as to enhance accessibility. We also provide additional model simulations to enhance interpretability (p. 19/20, subheading “Inter-individual differences in drug effects” and see Appendix 1 – Figure 10).

5. Reviewers noted the lack of effects of levodopa on MF and MB learning. While this is something that has been seen previously, it also might reflect lack of sensitivity of the manipulation with the current task. It would be important to elaborate on the implications of the lack of basic MF and MB modulation.

Regarding a lack of a drug effect on MF learning or control, we now elaborate on this on p. 22/23:

“With respect to our current task, and an established two-step task designed to dissociate MF and MB influences (Daw et al., 2011), there is as yet no compelling evidence for an direct impact of DA on MF learning or control (Deserno et al., 2015a; Kroemer et al., 2019; Sharp et al., 2016; Wunderlich et al., 2012, Kroemer et al., 2019). […] DA tone is proposed to encode average environmental reward rate (Mohebi et al., 2019; Niv et al., 2007), a putative environmental summary statistic that might in turn impact an arbitration between behavioural control strategies according to environmental demands (Cools, 2019).”

As pointed out by the reviewers as well, in the present task we did not find an effect of levodopa on MB influences per se and now discuss this on p. 22:

“In this context, a primary drug effect on prefrontal DA might result in a boosting of purely MB influences. […] this regard, our data is suggestive of between-subject heterogeneity in the effects of boosting DA on distinct aspects of MB influences.”

In response to the reviewers, we have now simplified this analysis of inter-individual differences and extend upon it by also providing model simulations (see response to point 4).

An open question remains as to why different task conditions (guidance of MFCA by MB vs. pure MB control) apparently differ in their sensitivity to the drug manipulation. We discuss this (p. 22) by proposing that a cost-benefit trade-off might play an important role (Westbrook et al., 2020).

6. Relatedly, the critical MB guidance of MFCA effect was not observed under placebo (Figure 3 B). The computational model-based analyses do speak to this issue, but are the results from the model-informed and model-agnostic analyses otherwise consistent? Model-agnostic analyses reveal a greater effect of levodopa on informative destination for the GN than the GR trials and no effect for non-informative destination. Conversely model-informed analyses reveal a nomination effect of levodopa across informative and non-informative trials. This was not addressed. Also, how does this model space, the winning model and the modeling exercise differ (or not) from that in the previous paper by Moran et al. without levodopa?

Thank you for the opportunity to discuss discrepancies between our model-agnostic and computational modelling analyses. In brief, the results from the computational model are generally statistically stronger, which is not surprising given that they are based on influences from far more trials. Further, although the computational model uses a slightly different parameterization from that reported in Moran (2019), it is a formal extension of that model, allowing the strength of effects for informative and uninformative destinations to differ.

To amplify: firstly, in contrast to Moran et al., 2019, the model-agnostic contrast in repeat vs. switch trials, providing a measure of preferential guidance of MF choice repetition, was not significant under placebo [p. 13: “…but not under placebo (b=-.02, F(239,1)=.52, p=.474)”; Moran et al. 2019: p = 0.028]. However, as pointed out by the editor, the corresponding effect reached (one-sided) significance based on the parameters of the computational model [p. 19: “…was only marginally significant in the placebo condition (b = 0.05, F(1,240) = 2.77, one-sided p=.048)”].

Secondly, we did not find a drug enhancing effect on guidance of MFCA based on reward at the non-informative destinations as assessed in clash trials condition [p. 16: “The interaction effect between drug and non-informative reward, however, was not significant (b=0.05, t(4861)=.39, p=.696, Figure 4B2)”]. However, analysing the parameters of the computational model as reported showed a nomination x drug interaction [p. 18: “Crucially, we found a positive nomination x drug interaction (b=0.10, t(480)=2.43, p=.015).”]. We believe a critical difference here is that the model-agnostic analyses are performed separately on informative and non-informative destinations (of necessity, since they are measured in different ways), whereas the computational modelling integrates the two. Indeed, if we look separately at the computational modelling parameters for the different destinations, we find that the nomination x drug interaction were weakly (non-significantly) positive for both [informative: (b=.11, t(244)=1.93, p=.055); non-informative (b=.09, t(244)=1.72, p=.087)]. This is consistent with a lack of triple interaction in the analysis based on computational modelling [p. 18: “Importantly, this interaction was not qualified by a triple interaction, nomination x drug x informativeness (b=.02, t(480)=0.32, p=.738)”]. Thus, again, we consider this speaks to extra power of the computational modelling.

In sum, the reviewers are correct that the results (in terms of crossing the threshold of statistical significance) are not always consistent between the model-agnostic and the computational modelling approaches. Such discrepancies may be expected when effects of interest are weak to moderate, which we acknowledge (p. 25/26, limitations). We think in this regard the computational modelling analyses is superior in its sensitivity to capture effects of interest because this approach integrates information across multiple trials and trial-types. We now include a discussion of this in more detail in the section on limitations (p. 25/26).

With respect to computational models fitted to the choice data, while the reported models are conceptually very similar to the one’s reported in Moran et al., (2019), we indeed changed the model’s and sub-model’s parameterization to one involving credit assignment (ca) and forgetting (f) (where the rewards (r) are coded as ±1):

(Equation. 1)Q←(1−f)∗Q+ ca×r

rather than learning rate (α) and reward sensitivity (β; which is equivalent to inverse temperature).

(Equation. 2)Q←Q+α(βr−Q)

However, these are formally equivalent (setting =f ; and β=ca/f). We discussed in detail the similarities and differences between models based on learning-rate and credit assignment parameterization in a recent paper (Moran et al., 2021), to which we now refer the readers. The substantive change is to allow the MF updates on an uncertainty trial to have four different ca parameters – one for each combination of vehicle (ghost-nominated and ghost-rejected) and destination (non-informative [and first]; and informative [and second]). These extra parameters were justified according to our complexity measure for model selection. Our use of a variant of Equation 1 allows a more elegant means to have two credit assignment parameters per update than would have been possible with a variant of Equation 2. We now include a reference to this change in parameterization in the limitation section (p. 25/26), and include a more detailed description in Appendix 1 (p. 45-47).

Finally, to test if the current models support our main conclusion from Moran et al. (2019) that retrospective MB inference guides MFCA for both the informative and non-informative destinations, we reanalysed the Moran et al. data using the current novel models and found converging support, as we now report (Appendix 1 – Figure 8).

7. What putative neural mechanism led the authors to predict this selective modulation of the interaction? The introduction states that "Given DA's contribution to both MF and MB systems, we set out to examine whether this aspect of MB-MF cooperation is subject to DA influence." This is vague. For the hypothesis to be plausible, it would need to be grounded in some idea about how the effect would be implemented. Where exactly does dopamine act to elicit an effect on retroactive MB inference, but not MB learning? If the mechanism is a modulation of working memory and/or replay itself, then shouldn't that lead to boosting of both MB learning as well as MB influences on MF learning? Addressing this involves specification of the mechanistic basis of the hypothesis in the introduction, but the question also pertains to the Discussion section. Hippocampal replay is invoked, but can the authors clarify why a prefrontal working memory (retrieval) mechanism invoked in the preceding paragraph would not suffice. In any case, it seems that an effect of dopamine on replay would also alter MB planning.

We agree with this criticism and have now revised the relevant intro paragraph (p. 3/4). After pointing out some of the various ways that MB and MF learning interact, we discuss DAergic manipulation of replay in particular (p. 24). We infer that a component of a MB influence over choice comes from the way it trains a putative MF system (something explicitly modelled in Mattar and Daw, 2018, and a new preprint from Antonov et al., 2021, referencing data from Eldar et al., 2020) – and consider what happens if this is boosted by DA manipulations. The difference between the standard two-step task and the present task is that in our task there is extra work for the MB system in order to perform inference so as to resolve uncertainty for MFCA. We later suggest that the anticorrelation we found between the effect of DA on MB influence over choice and MB guidance of MFCA arises from this extra work.

The broader questions that the editor raises about (prefrontal) working memory and (hippocampal) replay pertains to recent and ongoing work, and we feel this should be part of the discussion. However, as our previous Discussion section on these issues was also criticized as too speculative, we have re-written this to deal with the questions raised by the editor so as to detail more clearly different possible mechanistic explanations, pointing to how they might be tested in the future (p. 23/24).

8. The critical drug effect seems marginally significant and the key plots in Figure 3B and Figure 4 B, C and D do not show relevant variability in the drug effect. It would be important to connect (or otherwise plot) individual data points between the levodopa and placebo conditions, allowing readers to appreciate the relevant individual variability in the drug effects.

We agree and have now replotted the data in the new Figure 4A3 and B2, as well as Figure 5 B, C and D to reflect drug-related variability. With regard to the initial version of new Figure 4 (Figure 3 previously), we consider plotting individual variability with respect to task conditions is also of interest and now include this in Appendix 1 (Appendix 1 – Figure 4).

9. Please provide a rationale for changing the stimuli and details of the task relative to what was used in Moran et al.

Based on previous data collection with this task (Moran et al., 2019), we reasoned that a less arbitrary connection and a natural meaning (vehicles and destination as compared of objects and colours) would make the cover story more plausible and entertaining and, thus, could facilitate compliance in pharmacological within-subjects study. Further, for vehicle-destination mappings there is a broad range of stimuli available for two version of the task. The only further change was that the task was slightly shortened from 504 (7 blocks with 72 trials) to 360 (5 blocks with 72) trials per session (720 trials across drug and placebo trials). This latter decision was made based on the data by Moran et al., which indicated comparable results when cutting down to 360 trials and was preferable for the pharmacological design. No other changes were implemented to the task. We now refer to this in the methods section on p. 28.

10. The amount of training depended on performance on a quiz. How much training did people receive and was this matched between drug sessions?

At initial training, participants first saw one vehicle and had to press the space bar, in a self-paced manner, to subsequently visit the two associated destinations in random order. There were 12 repetitions per vehicle-destination mapping (48 trials). Following this initial training, participants responded to two types of quiz trials where they were either asked to match one destination out of two to a vehicle, or to match one vehicle out of two to a destination within a time limit of 3sec (8 trials per quiz type). To ensure each and every participant had the same level of knowledge of the transition structure, each quiz trial had to be answered correctly and in time (<3s) within the placebo and drug sessions (which had different sets of stimuli). Otherwise, a further training session followed but now with only 4 repetitions per vehicle-destination mapping (16 trials), followed again by the two type of quiz trials. This was repeated until criterion was reached. The criterion was identical for each participant in placebo and drug sessions, while the number of training cycles until criterion was reached could vary. The average number of training cycles (drug: mean 3.2381, std 3.0518, min 1, max 18; placebo: mean 3.6774, std 2.7328, min 1, max 13) did not differ between sessions (Wilcoxon signed rank test, p=.1942).

Participants also received further written instructions on trial types (see Methods, p. 29). Before starting the actual main experiment, subjects were required to perform a short refresher training of the vehicle-destination mappings (with 4 repetitions per vehicle-destination mapping), followed by a requirement to pass the same quiz trials as described above. If they failed to pass at this stage, the refresher training was repeated with 2 repetitions per vehicle-destination mapping until such time as the quiz was passed. The average cycles of refresher training (drug: mean 2.2222, std 1.5600, min 1, max 10; placebo: mean 2.0476, std 1.4528, min 1, max 9) did not differ between sessions based on a Wilcoxon signed rank test (p=.3350).

This additional information on training and training cycles is now included in the Methods section (p. 28/29).

11. Please include a note on the implications of this study to psychiatric disorders that are related to dopamine.

We now include a consideration of implications for psychiatric symptoms (p. 24/25):

“MB influences have previously been studied in relation to psychiatric symptom expression (Deserno et al., 2015a; Gillan et al., 2016; Voon et al., 2015). […] Moreover, emerging evidence for a positive influence of higher DA levels on MB influences, and as revealed in our study a differential sensitivity of distinct MB process to enhancing DA levels, provides a potential target to dissect variability in treatment responses to therapeutic agents which boost DA levels (for example, psychostimulants in ADHD).”

12. In addition, individual reviewers made a number of suggestions to improve the manuscript. Please consider these additional points when preparing the revised manuscript.

We address additional points as follows if they have not already been covered by our responses to the Reviewing Editor’s summary.

Reviewer #1 (Recommendations for the authors):1. It would be important to expand the discussion of two somewhat surprising findings. First, please expand on why individual differences in the effect of dopamine on MB inference would negatively correlate with the effects of retrospective MB inference on MFCA. Second, please elaborate on why you think the current results do not replicate previously reported effects of dopamine on MB learning.

We follow this suggestion and have expanded the discussion, including addressing the negative correlation between inter-individual differences in drug effects on p. 21/22 (the analysis of which have been simplified, see response to point 4 to the editor’s summary). The second issue with regard to replication is covered in response to point 5 by the Editor.

2. The logic of the analysis is complex. It took me a while to figure out what the critical effects are Figure 3. You may want to consider revising the presentation of their analyses, if possible. Could you expand the schematics in the figures? Perhaps you could add a flow chart of how the different effects are isolated. I realize it may be difficult to come up with this, but I believe it would make your findings much more accessible.

Thank you for this prompt. We have considerably amended the presentation of the task’s logic and analyses, as pointed out in more detail in response to point 3 by the Editor. We include flow charts to Figures 2 and 3 to illustrate MF vs. MB modelling and trial types.

Reviewer #2 (Recommendations for the authors):– I believe that the most important effect in this study is the one presented in Figure 3C and 3D, which authors have called the "Clash condition". In this one, the same chosen pair on the preceding uncertainty trial is presented in a standard trial and subjects are asked to choose between the two choices. This is, I believe, is the ultimate test trials in the study; and there is no significant effect of drug in those trials. Looking at Table S2, it seems to me that authors have done a very good job in increasing the within-subject power for that trial type (mixed-effect df for the clash trials is 4861; the df for the repeat/switch trials is 239 according to Table S2). Related to this point, authors have found no significant effect of DA on credit assignment in their computational modeling analysis.If this is correct (and I am happy to be convinced otherwise), then I have two suggestions. First, authors should do a Bayesian test to test if the null hypothesis (no drug effect) is significantly preferred to the alternative hypothesis. If this is the case, I think the study is very informative (but actually in the opposite direction of the current claim). Second, if the Bayesian test does not prefer the null hypothesis, then I suggest to revise the title/abstract and conclusion part of the study; and simply state that there was no conclusive effect of DA on credit assignment. In this case, I think the other part of results (i.e Figure 3B) as well as the span-dependent effect on parameters from the model-based analyses (Figure 5) are of interest and significance.

We have replied to some of these issues in response to point 1, as summarized by the editor. In short, the conditions are designed to examine different aspects of how credit is assigned in the MF system based on MB guidance. For the informative destination, the conditions examine credit assignment to the ghost-nominated (“repeat standard trials n+1 following uncertainty trials n”) and ghost-rejected (“switch standard trials n+1 following uncertainty trials n”) vehicle, and the contrast between repeat and switch standard trials reflects preferential MFCA – the first variable of interest. The “clash condition” examines credit assignment to the ghost-selected vehicle based on the non-informative destination – the second variable of interest. Thus, the conditions capture different aspects of MFCA as guided by MB inference, but they do not have different levels of relevance to isolate retrospective MB inference.

With respect to rewards at the non-informative destination, we did not find a significant common-reward x drug interaction in the model-agnostic analysis of clash standard trials n+1. However, and in contrast to the reviewers’ comment, we did find evidence that MFCA based on the non-informative destination is enhanced under levodopa based on the computational modelling. In response to point 6, as raised by the editor, we address apparent discrepancies between model-agnostic and modelling-derived analyses more fully. Lastly, we have amended our description of the logic for the different conditions to make the distinction between trial conditions more clear.

– The part of discussion about the hippocampal replay comes across as highly speculative and not grounded in the current findings. I suggest to instead focus on limitations of the current study.

We understand that this paragraph came across rather speculative. However, in response to point 7 raised by the editor, we were asked to be more specific about the potential neural mechanisms at play. We now include discussion of previous neural findings that substantiate our hypotheses in the introduction (p. 3/4). We have re-written the criticized paragraph in the discussion to be less speculative (p. 23/24). Thus we follow the suggestion of the editor to discuss which DA-dependent behaviour measured in our task might depend on working memory as compared to replay. Furthermore, we have also expanded the discussion of the limitations of our study (p. 25/26).

– Labels are not matched with panels in Figure 4.

We apologize for this mistake, which is now corrected.

– I suggest to move Figure 2 to the supplementary, or at least present it after Figure 3. This figure is not about the credit assignment, which is the main focus of this study.

We respectfully disagree and would ask the referee to reconsider. Figure 2 does capture MF and MB credit assignment, but note this is in the context of standard trials rather than uncertainty trials. We intend Figure 2 to show MF and MB influences for drug and placebo (which is itself important) and also to help in setting up the logic of MF and MB influences (that we then exploit for the uncertainty trials too). Thus, we think removing this figure would diminish the accessibility of the results for a readership that is not familiar with the type of task design we have adopted.

– I don't think that authors statement of Cools' theory in terms of DA synthesis capacity is correct (and it is a bit unfair too) (page 17). I believe the main prediction, based on that theory, or at least its recent form, is that DA effect is baseline-dependent and therefore it is actually quite consistent with what authors found. Based on this theory, WM-span test is a good "proxy" of baseline dopamine synthesis capacity. I suggest to revise that part of the discussion.

We apologize for a lack of clarity in this regard. Indeed, we think that our findings are entirely compatible with the assumption that DA drug effects are baseline-dependent. Also, our revised analysis of inter-individual differences in drug effects indicate substantial variability, something we speculate could be accounted for by differential baseline DA levels. An interesting point for future work will be to test, based on this theory, whether low versus high baseline levels of DA also account for this drug-related heterogeneity as seen with respect to increasing guidance of MF learning by MB inference, in contrast to pure MB evaluations under enhanced DA levels. This would also be expected under an assumption that MB guidance of MF and MB evaluations indeed have different cost-benefit characteristics. This can be seen as in line with recent findings implicating baseline DA levels in moderating DA drug effects on cost-benefits arbitration (Westbrook et al., 2020). The moderation of our drug effect finding by WM span is not fully in line with this idea. However, we have opted to refrain from discussing WM as a proxy for baseline DA not least following the recommendation by the editor and other reviewer that WM findings should be removed from the main text (now in Appendix 1).

– I believe that a note on how this study might be related to psychiatric disorders that are related to dopamine would be of interest for many readers.

This was now included to the discussion (also compare point 11 by the Editor). We now write in the discussion (p. 24/25):

“MB influences have previously been studied in relation to psychiatric symptom expression (Deserno et al., 2015a; Gillan et al., 2016; Voon et al., 2015). […] Moreover, emerging evidence for a positive influence of higher DA levels on MB influences, and as revealed in our study a differential sensitivity of distinct MB process to enhancing DA levels, provides a potential target to dissect variability in treatment responses to therapeutic agents which boost DA levels (for example, psychostimulants in ADHD).”

Reviewer #3 (Recommendations for the authors):It would be good make explicit exactly how many trials there were for the various conditions of interest: e.g. MF choice repetition trials, MB choice generalization trials and critical retrospective MB inference trials.

This is now provided (p. 30). In response to point 2 by the editor, we write:

Each session (drug as well as placebo) had 360 trials, resulting in 720 trials in total. One third of trials (120 per session) were standard trials (following standard trials) and included 30 presentations of each of four eligible pairs of vehicles in random order. Half of these trials are used for each of the analyses of MF and MB signatures (Figure 2). This resulted in a maximum of 60 trials for the analysis of each of the MF and MB signatures. For each session, every third trial was an uncertainty trial (120 trials), which were always followed by a standard trial, where a third of these trials (40 trials) contributed to each of the “repeat”, “switch” and “clash” conditions (now Figure 4). Accounting for missed choices, this resulted in the following trial averages per condition and per session (placebo/drug): MF-placebo 58.07, MF-drug 59.15; MB-placebo 58.00, MB-drug 57.01; repeat-placebo 39.42, repeat-drug 39.31; switch-placebo 39.34, switch-drug 39.10; clash-placebo 39.34, clash-drug 39.32.

Figure 3C: is there a mistake in the destinations paired with the yellow car on the standard trial n+1?

We apologize for this mistake, which is now corrected.

Please provide a rationale for changing the stimuli and details concerning shortening the task cf Moran et al.

We responded to this in reply to point 9 by the editor as follows:

Based on previous data collection with this task (Moran et al., 2019), we reasoned that a less arbitrary connection and a natural meaning (vehicles and destination as compared of objects and colors) would make the cover story more plausible and entertaining and, thus, could facilitate compliance in pharmacological within-subjects study. Further, for vehicle-destination mappings there is a broad range of stimuli available for two version of the task. The only further change was that the task was slightly shortened from 504 (7 blocks with 72 trials) to 360 (5 blocks with 72) trials per session (720 trials across drug and placebo trials). This latter decision was made based on the data by Moran et al., which indicated comparable results when cutting down to 360 trials and was preferable for the pharmacological design. No other changes were implemented to the task. We now refer to this in the methods section on p. 28.

The amount of training depended on performance on a quiz. How much training did people receive and was this matched between drug sessions?

This is also part of our response to point 10 by the editor:

At initial training, participants first saw one vehicle and had to press the space bar, in a self-paced manner, to subsequently visit the two associated destinations in random order. There were 12 repetitions per vehicle-destination mapping (48 trials). Following this initial training, participants responded to two types of quiz trials where they were either asked to match one destination out of two to a vehicle, or to match one vehicle out of two to a destination within a time limit of 3sec (8 trials per quiz type). To ensure each and every participant had the same level of knowledge of the transition structure, each quiz trial had to be answered correctly and in time (<3s) within the placebo and drug sessions (which had different sets of stimuli). Otherwise, a further training session followed but now with only 4 repetitions per vehicle-destination mapping (16 trials), followed again by the two type of quiz trials. This was repeated until criterion was reached. The criterion was identical for each participant in placebo and drug sessions, while the number of training cycles until criterion was reached could vary. The average number of training cycles (drug: mean 3.2381, std 3.0518, min 1, max 18; placebo: mean 3.6774, std 2.7328, min 1, max 13) did not differ between sessions (Wilcoxon signed rank test, p=.1942).

Participants also received further written instructions on trial types (see Methods, p. 29). Before starting the actual main experiment, subjects were required to perform a short refresher training of the vehicle-destination mappings (with 4 repetitions per vehicle-destination mapping), followed by a requirement to pass the same quiz trials as described above. If they failed to pass at this stage, the refresher training was repeated with 2 repetitions per vehicle-destination mapping until such time as the quiz was passed. The average cycles of refresher training (drug: mean 2.2222, std 1.5600, min 1, max 10; placebo: mean 2.0476, std 1.4528, min 1, max 9) did not differ between sessions based on a Wilcoxon signed rank test (p=.3350).

This additional information on training and training cycles is now included in the Methods section (p. 28/29).

While I consider myself somewhat of an expert, I found the paper really quite difficult to parse, in part due to the large number of abbreviations, constructs and interactions invoked. So I am somewhat concerned that it will also be hard to appreciate for a somewhat broader audience. One thing that can be done is to be more consistent in the use of relevant labels:For example, ghost trials vs uncertainty trials, PMBCA vs retrospective MB inference

We appreciate this concern and have amended several changes to enhance accessibility of the manuscript. These changes are outlined in response to point 3, as summarized by the editor. We have also carefully worked on the manuscript to improve consistency with regard to abbreviations and labels.

References:

Collins, A.G.E., and Frank, M.J. (2014). Opponent actor learning (OpAL): Modeling interactive effects of striatal dopamine on reinforcement learning and choice incentive. Psychological Review 121, 337–366.

Cools, R. (2019). Chemistry of the Adaptive Mind: Lessons from Dopamine. Neuron 104, 113–131.

Dayan, P. (2009). Dopamine, reinforcement learning, and addiction. Pharmacopsychiatry 42 Suppl 1, S56-65.

Deserno, L., Huys, Q., Boehme, R., Buchert, R., Heinze, H.J., Grace, A.A., Dolan, R.J., Heinz, A., and Schlagenhauf, F. (2015a). Ventral striatal presynaptic dopamine reflects behavioral and neural signatures of model-based control during sequential decision-making. Proc Natl Acad Sci U S A.

Deserno, L., Wilbertz, T., Reiter, A., Horstmann, A., Neumann, J., Villringer, A., Heinze, H.J., and Schlagenhauf, F. (2015b). Lateral prefrontal model-based signatures are reduced in healthy individuals with high trait impulsivity. Transl Psychiatry 5, e659.

Doll, B.B., Jacobs, W.J., Sanfey, A.G., and Frank, M.J. (2009). Instructional control of reinforcement learning: a behavioral and neurocomputational investigation. Brain Res 1299, 74–94.

Doll, B.B., Hutchison, K.E., and Frank, M.J. (2011). Dopaminergic genes predict individual differences in susceptibility to confirmation bias. J Neurosci 31, 6188–6198.

Doll, B.B., Bath, K.G., Daw, N.D., and Frank, M.J. (2016). Variability in Dopamine Genes Dissociates Model-Based and Model-Free Reinforcement Learning. J Neurosci 36, 1211–1222.

Everitt, B.J., and Robbins, T.W. (2005). Neural systems of reinforcement for drug addiction: from actions to habits to compulsion. Nat Neurosci 8, 1481–1489.

Gillan, C.M., Kosinski, M., Whelan, R., Phelps, E.A., and Daw, N.D. (2016). Characterizing a psychiatric symptom dimension related to deficits in goal-directed control. *eLife* 5.

Groman, S.M., Massi, B., Mathias, S.R., Curry, D.W., Lee, D., and Taylor, J.R. (2019). Neurochemical and Behavioral Dissections of Decision-Making in a Rodent Multistage Task. J Neurosci 39, 295–306.

Hogarth, L. (2020). Addiction is driven by excessive goal-directed drug choice under negative affect: translational critique of habit and compulsion theory. Neuropsychopharmacol. 45, 720–735.

Kroemer, N.B., Lee, Y., Pooseh, S., Eppinger, B., Goschke, T., and Smolka, M.N. (2019). L-DOPA reduces model-free control of behavior by attenuating the transfer of value to action. NeuroImage 186, 113–125.

Kumakura, Y., and Cumming, P. (2009). PET studies of cerebral levodopa metabolism: a review of clinical findings and modeling approaches. Neuroscientist 15, 635–650.

Mohebi, A., Pettibone, J.R., Hamid, A.A., Wong, J.-M.T., Vinson, L.T., Patriarchi, T., Tian, L., Kennedy, R.T., and Berke, J.D. (2019). Dissociable dopamine dynamics for learning and motivation. Nature 570, 65–70.

Moran, R., Keramati, M., Dayan, P., and Dolan, R.J. (2019). Retrospective model-based inference guides model-free credit assignment. Nat Commun 10, 750.

Moran, R., Dayan, P., and Dolan, R.J. (2021). Human subjects exploit a cognitive map for credit assignment. Proc Natl Acad Sci USA 118, e2016884118.

Niv, Y., Daw, N.D., Joel, D., and Dayan, P. (2007). Tonic dopamine: opportunity costs and the control of response vigor. Psychopharmacology (Berl) 191, 507–520.

Pessiglione, M., Seymour, B., Flandin, G., Dolan, R.J., and Frith, C.D. (2006). Dopamine-dependent prediction errors underpin reward-seeking behaviour in humans. Nature 442, 1042–1045.

Redish, A.D. (2004). Addiction as a computational process gone awry. Science 306, 1944–1947.

Sharp, M.E., Foerde, K., Daw, N.D., and Shohamy, D. (2016). Dopamine selectively remediates “model-based” reward learning: a computational approach. Brain 139, 355–364.

Sharpe, M.J., Chang, C.Y., Liu, M.A., Batchelor, H.M., Mueller, L.E., Jones, J.L., Niv, Y., and Schoenbaum, G. (2017). Dopamine transients are sufficient and necessary for acquisition of model-based associations. Nat Neurosci 20, 735–742.

Simon, D.A., and Daw, N.D. (2012). Dual-System Learning Models and Drugs of Abuse. In Computational Neuroscience of Drug Addiction, B. Gutkin, and S.H. Ahmed, eds. (New York, NY: Springer New York), pp. 145–161.

Voon, V., Derbyshire, K., Ruck, C., Irvine, M.A., Worbe, Y., Enander, J., Schreiber, L.R., Gillan, C., Fineberg, N.A., Sahakian, B.J., et al. (2015). Disorders of compulsivity: a common bias towards learning habits. Mol Psychiatry.

Westbrook, A., van den Bosch, R., Määttä, J.I., Hofmans, L., Papadopetraki, D., Cools, R., and Frank, M.J. (2020). Dopamine promotes cognitive effort by biasing the benefits versus costs of cognitive work. Science 367, 1362–1366.

Wunderlich, K., Smittenaar, P., and Dolan, R.J. (2012). Dopamine enhances model-based over model-free choice behavior. Neuron 75, 418–424.

[Editors' note: further revisions were suggested prior to acceptance, as described below.]

Reviewer #2 (Recommendations for the authors):In this revision, Deserno and colleagues have revised the original manuscript extensively, especially regarding presentation of the task, analyses and findings. And I am thankful to authors for doing that.I still think that the clash condition is the more intuitive measure for CA in this task. But I am fine with the way that authors present and interpret their findings. One pressing issue, however, is still unresolved for me and I hope that authors can clarify that. In the previous revision, I asked about number of trials went into each condition. The main reason I asked that was because the df for the switch and repeat trials was very small given for a mixed effect within-subject analysis (i.e. 239, page 14). By examining the manuscript more thoroughly, I believe that the main reason is that authors have conducted a different type of analysis for those trials compared to the rest of manuscript. In particular, analysis of repeat and switch trials are based on proportion of choice (page 33.) This is quite different from the other analyses. What is the rationale for not doing a logistic regression here? Does a logistic regression with the same factors result in similar results? Proportion of binary dependent variable is not a good measure statistically. I might miss something here, but I believe clarifying this point makes the main results more convincing.

The reviewer raises an important and astute point. This has motivated us to reconsider our mixed-effect models. There are 2 issues. The first pertains to degrees of freedom; the second to the precise analysis type of repeat vs. switch trials. We address these issues in turn.

We appreciate the reviewer’s close inspection of the degrees of freedom. Upon reflection, we realized that in most of our models, our predictors are binary rather than continuous (with the exception of the model pertaining to MB choice on standard trials which includes the trial-by-trial changing reward probability as a continuous predictor, see Methods, p. 32, lines 785-794). Thus, there is only a limited number of predictor combinations. For example, if a model includes 3 binary predictors, then there are at most 2^3 = 8 predictor-combinations (there might be fewer if some combinations did not occur). In our previous analyses, each trial was described as a row (or observation) in the model. However, in cases where all predictors and the responses are binary, a better statistical practice is to aggregate across all repetitions of a predictor combination into a single row in the model. Each row is then characterised by the number of repetitions (i.e., trials that share this predictor combination) and the “number of successes” (i.e., the occurrence of an event of interest e.g., choice repetition). In this manner the model is formulated as a binomial (rather than binary) mixed-effects model. The logistic regression model is a special case of binomial regression when each predictor-combination occurs once. For example, if a participant contributes 100 trials for an analysis of interest, then in the logistic approach, he/she will contribute 100 rows, whereas in the binomial approach — at most there are 8 rows. Consequently, the binomial modelling approach yields lower degrees of freedom, and is more conservative. In fact, we have been using this approach in a recent study (Moran et al., 2021) and we apologize for only now incorporating this. We have revised all models reported in the manuscript and the respective tables in Appendix 1 (except for the model of MB choice in standard trials). While the degrees of freedom reduced as indicated, the differences in test-statistics beta, t and p values are tiny and the revised results reassuringly support the exact same conclusion.

With respect to the specific mixed-effects models on repeat vs. switch trials, the reviewer is indeed correct that one difference is that this linear model focuses on choice proportions, instead of on trial-by-trial choices. However, in our revised analysis of binomial mixed-effects models, all models analyse choice proportions (whenever only binary predictors are included and the response is binary).

However, another difference in our analysis of repeat/switch trials is that it relies on *reward-effects* i.e., the difference between choice proportions when the informative destination is rewarded *minus* unrewarded. In contrast, in the other (now binomial) models, we do not regress reward-effects but rather choice proportions where reward events (i.e., reward or non-reward) are included as a predictor. Relatedly, because reward-effects can be negative our repeat/switch model is linear, whereas the other models are binomial. Another subtle difference is that this “repeat/switch” analysis controls for rewards at the non-informative destination by directly averaging reward-effects (for the informative destination) across the cases that the non-informative destination was rewarded or unrewarded and therefore the non-informative reward does not need to be included as a regressor.

We have edited the main manuscript to take account of the above and to improve clarity on the issues raised.

Results, p. 14, lines 324-331:

“For repeat trials, we defined reward-effects as the contrast between the proportion of choices (trial n+1) of the vehicle the ghost nominated (on the uncertainty trial n), when the informative destination was previously rewarded vs. unrewarded. […] In a linear mixed-effects model (see Methods and Appendix 1 – Table 2) where we regressed these reward effects, …”

Methods, p. 32/33, lines 795-808:

“The analysis of how retrospective MB inference preferentially guides MFCA focused on standard (repeat and switch) trials n+1 following uncertainty trials n. […] An alternative detailed analysis using a binomial mixed-effects model, which focuses on choice proportions as a function of whether the informative destination was previously rewarded or not (in contrast to reward effects), is reported in Appendix 1.”

The reason we perform this analysis of repeat/switch trials based on reward-effects at the informative destination is that we aimed to keep the mixed-effects model simpler. The current model of reward- effects already includes 3 regressors (drug, nomination: switch/repeat and order; also see Results p. 14, lines 322-338; Methods p. 32/33, lines 795-808). A binomial mixed-effects model of choice proportions would include 5 regressors: the former 3 + reward/no-reward for informative destination + reward/non-reward for non-informative destination (as control). This results in an excessively complex 5-way model, which estimates many effects of no interest. Such complex models often do not converge and yield more noisy estimates for effects of interest (due to the very high number of estimated effects). Moreover, these models could potentially yield interactions of a very high order, which are difficult to interpret. We therefore chose to keep to our original analysis in the main text, which is also consistent with the analysis approach presented in the original task paper (Moran et al., 2019).

However, to fully address the reviewers’ remark and to test whether our conclusions were affected by the choice of analysis, we now ran the complex binomial mixed-effects model including data from repeat and switch trials. In this model, we regressed proportion of choices (rather than reward-effects) of a vehicle from the previously chosen vehicle-pair on predictors of interest including reward at the informative destination (I), trial-type (NOM) and drug but also rewards from the non-informative destination (N) and order of the drug session (ORDER) as controls with participants (PART) as random effects (this yields 5 predictors in total each with two levels coded as +.5 and -.5). Specifically, we formulate the model as:

CHOICE ~ 1 + (I * NOM * DRUG + N) * ORDER + (I * NOM * DRUG + N + ORDER | PART)

Importantly, we find a positive significant three-way informative-reward x nomination x drug interaction effect (b=0.39, t(973)=2.00, p=.046). This interaction was not qualified by a higher-order interaction. To interpret this interaction, we examined the simple two-way informative-reward x nomination interactions for L-Dopa and placebo. For L-Dopa, we found a positive two-way interaction (b = 0.42, F(1,973)=9.872, p=.002), which was explained by the fact that the simple informative reward-effect for repeat trials (b = 0.68, F(1,973)=46.06, p=2e-11) was stronger than for switch trials (b = 0.27, F(1,973)=8.46, p=.004). Thus, for L-Dopa we found preferential MFCA. In contrast, for placebo, the two-way informative-reward x nomination interaction was non-significant (b=.03, F(1,973)=.031, p=.859), providing no evidence for preferential MFCA. These results, now reported in Appendix 1 (p. 45/46, lines 1117-1139), support the very same conclusions as our analysis based on reward-effects reported in the main text. Finally, we note that running the same complex model in the binary/logistic trial-by-trial setup, also supports the same conclusions (informative-reward x nomination x drug interaction effect b=0.39, t(9726)=2.00, p=.046). Thus, reassuringly we conclude that our finding of enhanced preferential MFCA (under L-dopa as compared to placebo) based on rewards at the informative destination does not depend on the choice of analysis type.

References:

Moran, R., Keramati, M., Dayan, P., and Dolan, R.J. (2019). Retrospective model-based inference guides model-free credit assignment. Nat Commun 10, 750.

Moran, R., Dayan, P., and Dolan, R.J. (2021). Efficiency and prioritization of inference-based credit assignment. Current Biology S0960982221004644.